# Epigenetic mechanisms to propagate histone acetylation by p300/CBP

Masaki Kikuchi [1], Satoshi Morita[1], Masatoshi Wakamori[1], Shin Sato[1], Tomomi Uchikubo-Kamo[2], Takehiro Suzuki[3], Naoshi Dohmae [3], Mikako Shirouzu [2] & Takashi Umehara [1] ✉

Histone acetylation is important for the activation of gene transcription but little is known about its direct read/write mechanisms. Here, we report cryo-genic electron microscopy structures in which a p300/CREB-binding protein (CBP) multidomain monomer recognizes histone H4 N-terminal tail (NT) acetylation (ac) in a nucleosome and acetylates non-H4 histone NTs within the same nucleosome. p300/CBP not only recognized H4NTac via the bromodomain pocket responsible for reading, but also interacted with the DNA minor grooves via the outside of that pocket. This directed the catalytic center of p300/CBP to one of the non-H4 histone NTs. The primary target that p300 writes by reading H4NTac was H2BNT, and H2BNTac promoted H2A-H2B dissociation from the nucleosome. We propose a model in which p300/CBP replicates histone N-terminal tail acetylation within the H3-H4 tetramer to inherit epigenetic storage, and transcribes it from the H3-H4 tetramer to the H2B-H2A dimers to activate context-dependent gene transcription through local nucleosome destabilization.

In eukaryotes, genomic DNA is packaged within the cell nucleus by forming the nucleosome, the structural unit of chromatin[1]. The nucleosome, 11 nm in diameter and 5.5 nm high, consists of 145–147 base pairs (bp) of DNA wrapped around a histone octamer composed of one [H3-H4]$_2$ tetramer and two [H2A-H2B] dimers[2]. In the nucleosome, eight NTs of four histone pairs protrude outside of DNA. On the side chains of each of these NTs, various post-translational modifications (PTMs) are chemically written, read, or erased by diverse proteins[3,4]. A typical PTM is acetylation of the Nε nitrogen of the lysine (K) side chain of histone NTs, which correlates with transcriptional activation of genes in eukaryotes[5–7].

Representative writers of lysine acetylation (Kac) are E1A-binding protein p300 (EP300; KAT3B)[8] and its homolog, CBP (KAT3A)[9], which can transfer an acetyl group onto the NT lysine residue(s) for all four histones[10–12]. A typical reader of Kac is the bromodomain, a four-helix bundle that forms a pocket preferring Kac[13]. p300 and CBP are unique in that they also contain bromodomains that can read Kac, for example, acetylated K12 of H4 (H4K12ac)[14,15]. Indeed, p300 acetylates NT of H2A.Z, an evolutionarily conserved variant of H2A, through the bromodomain-mediated H4NTac reader activity[16]. Interestingly, H2BNT is acetylated only by p300/CBP[10,17–19], which is considered a genuine signature of active enhancers and their target promoters[20,21].

Although H3K27ac, a residue acetylated by p300/CBP, has been used as a marker for active promoters and enhancers[22], it is absent in many p300-enriched chromatin regions[23] and is dispensable for enhancer activity of gene transcription in mouse embryonic stem cells[24]. In contrast, possibly through transcription-coupled histone exchange, H2BNTac better correlates with enhancer activity than any other known chromatin marks, with RNA transcription found in 79% of all H2BNTac-positive regions[20,21]. Importantly, the acetyltransferase activity of p300 responsible for H2BNTac[19] is a key driver of rapid enhancer activation and is essential for promoting the recruitment of

[1]Laboratory for Epigenetics Drug Discovery, RIKEN Center for Biosystems Dynamics Research, 1-7-22 Suehiro-cho, Tsurumi, Yokohama 230-0045, Japan. [2]Laboratory for Protein Functional and Structural Biology, RIKEN Center for Biosystems Dynamics Research, 1-7-22 Suehiro-cho, Tsurumi, Yokohama 230-0045, Japan. [3]Biomolecular Characterization Unit, Technology Platform Division, RIKEN Center for Sustainable Resource Science, 2-1 Hirosawa, Wako, Saitama 351-0198, Japan. ✉e-mail: takashi.umehara@riken.jp

RNA polymerase II (RNAPII) at virtually all enhancers and enhancer-regulated genes[25]. However, knowledge of how p300/CBP acetylates H2BNT and possibly thus activates transcription from enhancers and their target promoters has remained elusive.

Crystal structures of the histone acetyltransferase domain (HAT)[26] and a multidomain encompassing bromodomain, a RING finger (RING), a plant homeodomain finger (PHD), and HAT (BRPH)[15,27] of p300 suggest regulatory mechanisms for the catalytic reaction of p300/CBP. Recently, cryogenic electron microscopy (cryo-EM) structure analysis of a catalytically inactive p300 multidomain complexed with the unmodified nucleosome core particle (NCP) was reported[28]. However, the molecular mechanism of how p300/CBP reads/writes histone acetylation in the nucleosome(s) is unknown.

Interestingly, the Kac reader bromodomain of p300 is known to preferentially bind to H4NTac containing both K12ac and K16ac (H4K12ac/K16ac) that are stably maintained on mitotic chromatin[15,29]. In this study, we report cryo-EM structures revealing how the p300/CBP multidomain involved in the read/write of histone acetylation recognizes H4NTac containing H4K12ac/K16ac and acetylates non-H4 histone NTs in the same nucleosome. In addition, using various pre-acetylated nucleosomes, we report the directionality in which the catalytically active p300 multidomain reads/writes histone acetylation within a single nucleosome. Based on our data, we propose a model in which p300/CBP ensures epigenetic inheritance and expression via intranucleosomal acetylation of the H3-H4 tetramer and H2B-H2A dimers, respectively.

## Results

### Dependence of histone NTac on prior H4NTac

To understand how p300/CBP reads/writes histone acetylation, we expressed human p300$_{BRPHZT}$ protein (residues 1048–1836), containing bromodomain, RING, and PHD (BRP), the catalytically active HAT domain with the autoinhibitory loop (AIL), ZZ, and the TAZ2 domain (BRPHZT; Fig. 1a, Supplementary Fig. 1a, b), using a baculovirus expression system. The p300$_{BRPHZT}$ construct, which lacks the N- and C-terminal regions, was designed based on reports suggesting that similar human p300 constructs (residues 1035–1830 and residues

965–1810) read/write histone acetylation in the nucleosomes[16,30]. Similar to known catalytically active p300 protein constructs[31–34], the purified p300$_{BRPHZT}$ protein was confirmed to be an autoacetylated form containing K1542ac, K1546ac, K1549ac, K1550ac, K1551ac, K1554ac, K1555ac, K1558ac, and K1560ac that depends on its own catalytic activity (Supplementary Fig. 1c, Supplementary Table 1). For the nucleosome, based on the fact that p300$_{BRP}$ preferentially binds to H4NT containing H4K12ac/K16ac[15], we reconstituted a nucleosome, hereafter referred to as H4acNuc, consisting of the histone octamer having H4K12ac/K16ac and 146-bp palindromic human α-satellite DNA with 17-bp linker DNA linked to either end (i.e., a 180-bp nucleosome having H4K12ac/K16ac; Supplementary Fig. 1d, e).

First, we examined by immunoblotting whether K12ac/K16ac in H4 facilitates p300 to acetylate the nucleosomal histone NTs and for which residues (Fig. 1b, Supplementary Fig. 2). Consistent with previous reports[10,17], p300$_{BRPHZT}$ acetylated at least one residue (e.g., H2AK5ac, H2BK16ac, H3K27ac, and H4K8ac) for all four histone NTs, including when the nucleosome was unmodified. Overall, it acetylated H2B and H3 NTs more rapidly in H4acNuc than the unmodified nucleosome at the one-minute timepoint. With H4K12ac/K16ac present, p300$_{BRPHZT}$-catalyzed acetylation increased most prominently in H2BNT, increased significantly in H3NT (except H3K18ac), and decreased at K5/K8 in H4NT (probably due to proximity caused by p300 binding to H4K12ac/K16ac). Because the measurements shown in Fig. 1b and Supplementary Fig. 2 are independent, there are some differences, such as the degree of the H4K12ac/K16ac-dependent increase in H2BK16ac. However, both results are consistent in terms of statistical significance.

Next, we performed mass spectrometry to comprehensively detect histone residues whose acetylation by p300$_{BRPHZT}$ is facilitated in the presence of H4K12ac/K16ac (Supplementary Fig. 3; Supplementary Table 2). The results showed that at the one-minute post-reaction timepoint, the presence of H4K12ac/K16ac increased the amount of protease-digested peptides containing multisite H2BNTac of K11ac, K12ac, K15ac, K16ac, or K20ac by 4.3- to 46-fold. The amount of protease-digested peptides containing multisite H3NTac of K14ac, K18ac, K23ac, or K27ac also increased 1.5- to 11-fold. These p300$_{BRPHZT}$-

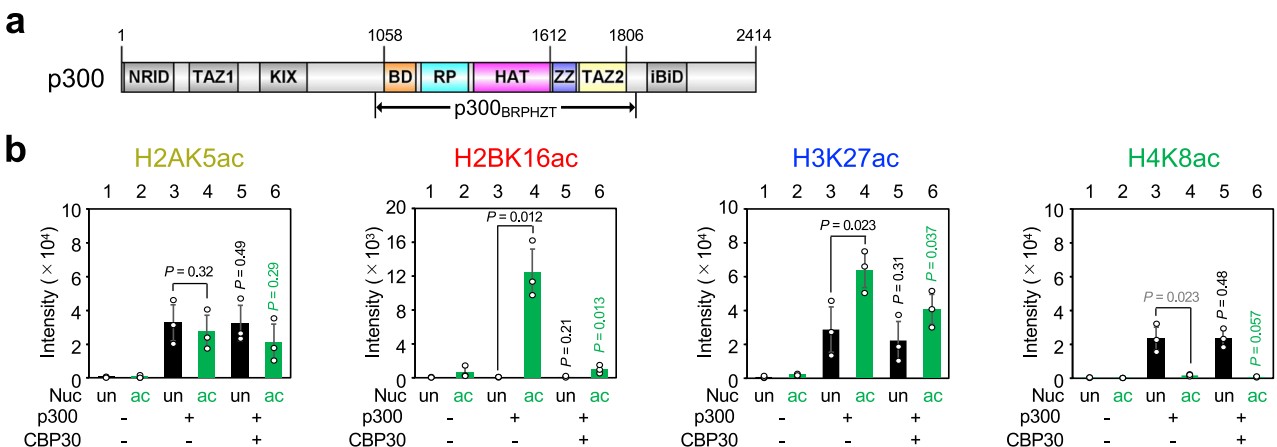

**Fig. 1 | Acetyltransferase activity of p300$_{BRPHZT}$ toward an H4-di-acetylated nucleosome. a** Schematic representation of the domain architecture of human p300. NRID nuclear receptor interaction domain, TAZ1 transcriptional adaptor zinc-finger domain 1, KIX kinase-inducible domain interacting domain, BD bromodomain, RP RING and PHD zinc-fingers, HAT histone acetyltransferase domain, ZZ ZZ-type zinc-finger, TAZ2 transcriptional adaptor zinc-finger domain 2, and IBiD IRF3-binding domain. The positions of the N- and C-termini and the start/end residues of the major domains are shown at the top. The positions of the start/end residues of the construct used in this study (i.e., p300$_{BRPHZT}$) are shown at the bottom. **b** In vitro acetyltransferase activity of p300$_{BRPHZT}$ toward an H4-di-acetylated nucleosome. The histone and residue for which acetylation was detected

by immunoblotting are shown above each panel. Color code: H2A, yellow; H2B, red; H3, blue; H4, green. Nucleosome (Nuc): un (black), unmodified; ac (green), H4K12/K16-acetylated. p300$_{BRPHZT}$ (p300): −, none; +, 1 μM. CBP30: −, none; +, 10 μM. CBP30 is an inhibitor that prevents the bromodomain pocket of p300 from binding to the acetylated histone N-terminal tails. The y-axis indicates the immunoblotting signal intensity at 1 min after the reaction. Data are mean ± standard deviation (SD) from three independent experiments. P value was calculated by a two-sample one-sided Welch's t-test. The alternative hypothesis is as follows: lane 4, increase vs. lane 3; lane 5, decrease vs. lane 3; lane 6, decrease vs. lane 4. P value shown in gray is not a significant increase.

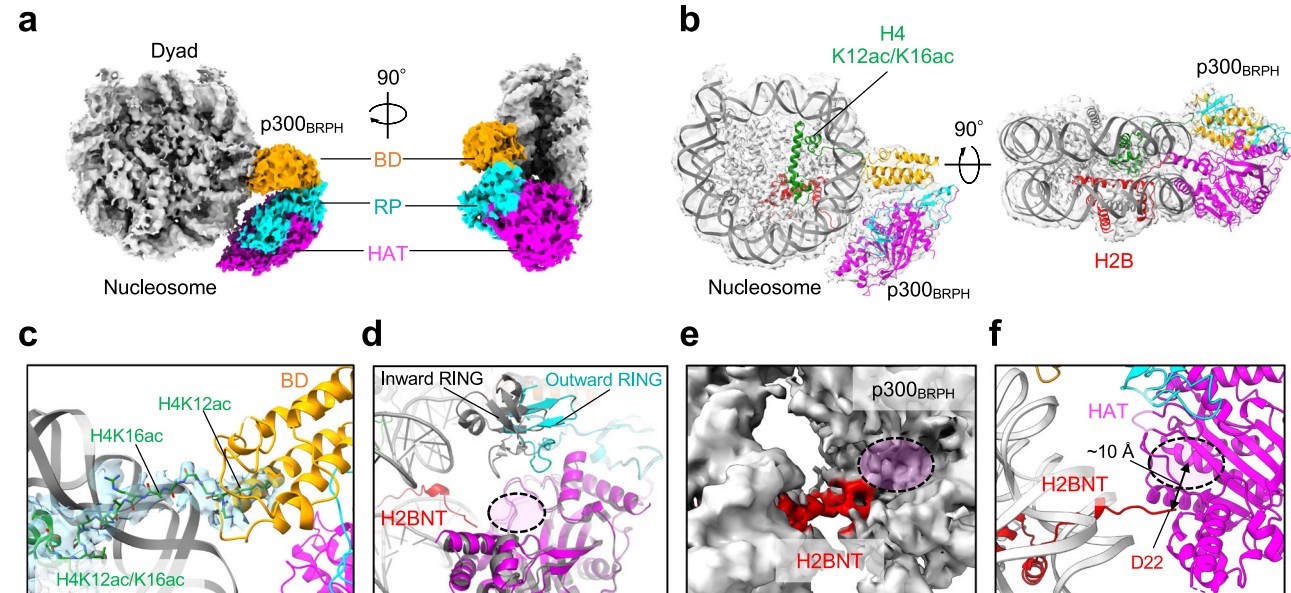

**Fig. 2 | Structure of p300$_{BRPHZT}$ bound to N-terminal tails (NT) in histone H2B and acetylated H4. a** Structure of p300$_{BRPH}$ bound to H2BNT and acetylated H4NT delineated by cryogenic electron microscopy (cryo-EM). Left, top view; right, side view. p300$_{BRPH}$ (#1 in Supplementary Fig. 9) binds to H4acNuc in a Slinky-like bent conformation via bromodomain and HAT. **b** Overall structure of p300$_{H2B}$ (#1) with H4-di-acetylated nucleosome in cartoon presentation. Color code: orange, p300 bromodomain (BD); cyan, p300 RING and PHD zinc-fingers (RP); magenta, p300 histone acetyltransferase domain (HAT); green, K12/K16-acetylated H4; red, H2B. **c** Close-up view of the binding mode of p300 bromodomain (BD, #1) to the H4-di-acetylated nucleosome (H4K12acK16ac). The map corresponding to H4NT is colored light blue. **d** Superposition of the cryo-EM structure of p300$_{BRPH}$ (#4) and the crystal structure of p300$_{BRPH}$ lacking AIL (PDB ID: 5LKU; gray). The magenta region circled in black is the substrate-binding site of HAT. **e** Close-up view of the cryo-EM map (#4) and the structure of H2BNT (red). **f** Close-up view of H2BNT (#4) shown as a cartoon representation.

catalyzed acetylation preferences were in good agreement with the results of immunoblot analysis. In the presence of CBP30[35], an inhibitor that prevents the bromodomain pocket of p300 from binding to the acetylated histone NTs, the p300$_{BRPHZT}$-catalyzed acetylation was selectively decreased at H2BK16ac and H3K27ac (Fig. 1b). These results suggest that when p300$_{BRPHZT}$ reads H4NTac at K12/K16, it writes Kac primarily on H2BNT and then H3NT.

## Structure of p300 bound to H4NTac and H2BNT
To elucidate the molecular mechanism by which p300/CBP reads/ writes histone acetylation of H4acNuc, we performed cryo-EM single particle structural analysis of catalytically active p300$_{BRPHZT}$. Our preliminary experiments using a 146-bp NCP containing H4K12ac/ K16ac yielded artificial structures in which p300$_{BRPHZT}$ bound across both ends of DNA (Supplementary Fig. 4a). To avoid this unnatural binding, we used nucleosome containing linker DNA. When the 180-bp nucleosome without H4K12ac/K16ac was used, we could not determine the structure of any of the obtained classes because the density resolution corresponding to p300$_{BRPHZT}$ was 8–12 Å (Supplementary Fig. 4b). The final three classes of p300 were each in close proximity to different histone NTs of the nucleosome and not to a specific histone NT. When the 180-bp nucleosome containing H4K12ac/K16ac (i.e., H4acNuc) was used, we obtained a group of structures in which p300$_{BRPHZT}$ binds to H4acNuc in several different modes by three-dimensional (3D) classification (Supplementary Figs. 5, 6). Of these complexes, the p300$_{BRPHZT}$·H4acNuc complex, in which HAT is in close proximity to H2BNT, had the largest number of single particles and its structure could be determined at the highest resolution (p300$_{H2B}$·H4acNuc; Fig. 2a, b). We obtained the 3D reconstruction density maps of p300$_{H2B}$·H4acNuc at 3.2–4.7 Å (Supplementary Table 3). In p300$_{BRPHZT}$, cryo-EM maps of BRPH were detected, but not of AIL, ZZ, or TAZ2. The structure-determined p300 multidomain (p300$_{BRPH}$) bound to H4acNuc in a Slinky-like bent conformation via bromodomain and HAT.

The bromodomain pocket of p300$_{BRPH}$ recognized the K12ac side chain of H4NT (Fig. 2c). RING of p300$_{BRPH}$ was structured in an outward-rotated conformation[15] (Fig. 2d), suggesting that HAT of p300$_{BRPH}$ in p300$_{H2B}$·H4acNuc is substrate-accessible. Around HAT of p300$_{BRPH}$, the density of H2BNT toward HAT could be modeled for the C-terminal residues after D22. Density maps were also detected near the substrate-binding pocket of HAT (Fig. 2e), but the residues of H2BNT bound to HAT could not be determined. The distance between the substrate-binding pocket and D22 (~10 Å) suggests that p300$_{BRPH}$ of this complex acetylates lysine residues from the N-terminal side of H2BNT up to K16 (Fig. 2f). Besides p300$_{H2B}$·H4acNuc, there were several similar complexes at low resolution in which H2BNT was located near the substrate-binding pocket of HAT (Supplementary Fig. 5g). This is consistent with our biochemical results that p300$_{BRPHZT}$ facilitated acetylation of multiple lysine residues from K11 to K20 of H2B when H4NT is pre-acetylated (Supplementary Fig. 3; Supplementary Table 2). Thus, it is likely that HAT of p300$_{BRPH}$ can acetylate various lysine residues around K16 of H2BNT by a similar structural mechanism. These results provide a structural basis for a read/write mechanism by which p300 recognizes H4NTac and acetylates H2BNT within the same nucleosome.

## Multiple modes of binding to histone NTs
Consistent with the fact that K12ac/K16ac in H4NT facilitated the p300$_{BRPHZT}$-catalyzed acetylation of multiple non-H4 histone NTs (Fig. 1b, Supplementary Figs. 2, 3), we obtained several cryo-EM structure classes in which HAT of p300$_{BRPHZT}$ bound with non-H4 histone NTs other than H2BNT in H4acNuc (Fig. 3a). In all classes, EM density was detected only for BRPH, as in p300$_{H2B}$·H4acNuc. We determined the complex structures in which HAT is directed toward H3NT (p300$_{H3-I}$·H4acNuc; Supplementary Table 3) or H2ANT (p300$_{H2A}$·H4acNuc). In all complexes, as in p300$_{H2B}$·H4acNuc, bromodomain of p300$_{BRPH}$ bound to H4K12ac of H4NTac, but the position of binding to the nucleosome was different for each

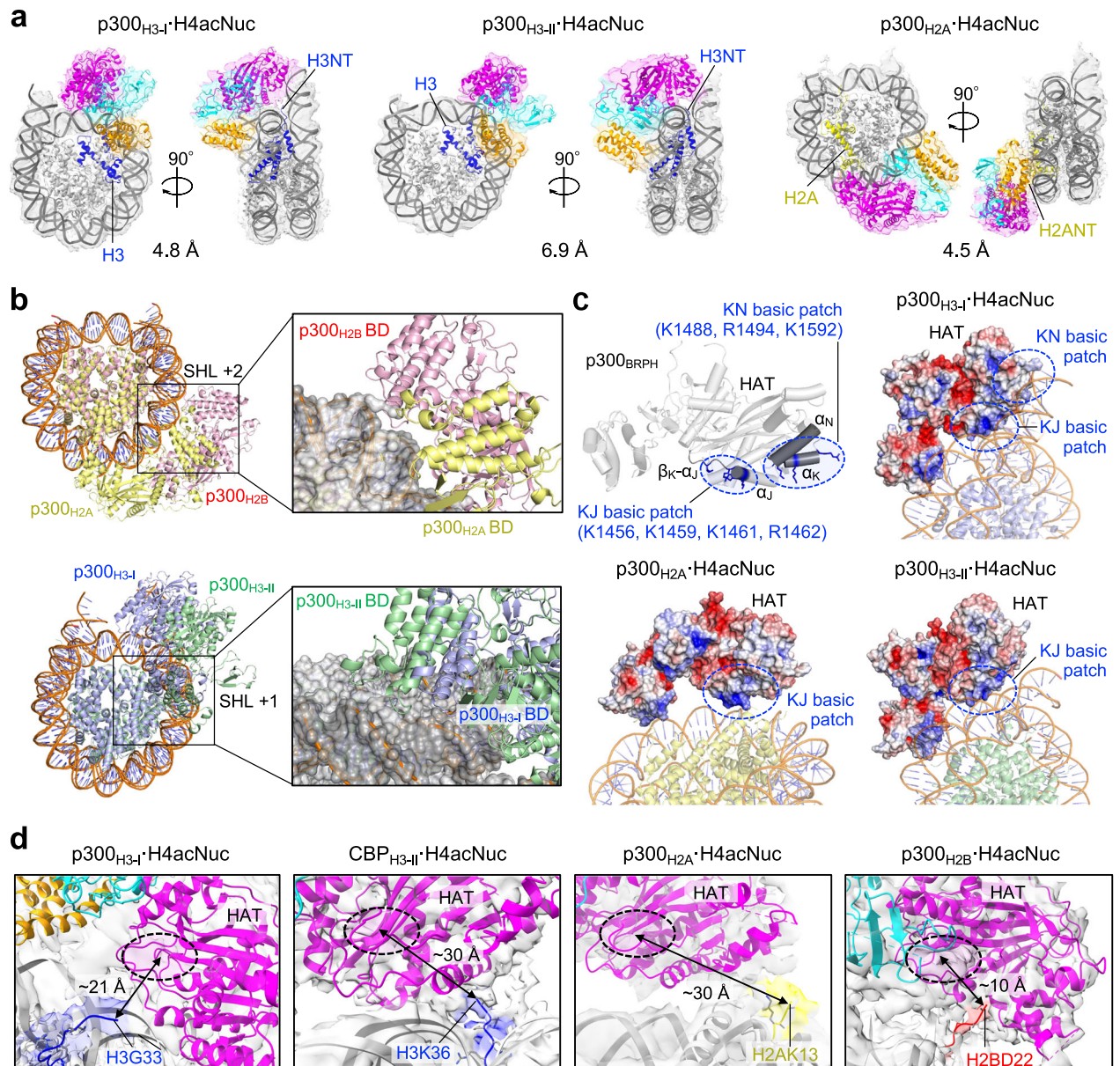

**Fig. 3 | Modes of binding to multiple histone N-terminal tails (NT). a** Various conformations of p300$_{BRPH}$ with the H4-di-acetylated nucleosome (H4acNuc) shown by cryogenic electron microscopy maps and structural modeling: left, p300$_{H3-I}$ (#5 in Supplementary Fig. 9); center, p300$_{H3-II}$ (#6); right, p300$_{H2A}$ (#7). (See Fig. 2 for color coding). **b** The positions of the superhelical location (SHL) at which p300 bromodomain (BD) interacts. Complex structures showing (top) superimposition of p300$_{H2B}$·H4acNuc (#4) and p300$_{H2A}$·H4acNuc (#7) and (bottom) superimposition of p300$_{H3-I}$·H4acNuc (#5) and p300$_{H3-II}$·H4acNuc (#6). The respective regions where p300 BD interacts with DNA are indicated by black squares and are shown on the right in close-up, displaying p300 (ribbon diagram) and nucleosome (surface diagram). **c** Basic patches interacting with DNA at p300 histone acetyltransferase domain (HAT). In the top left panel (#5), two basic patches are circled in blue. One basic patch (K1456, K1459, K1461, and R1462) is located

around the $\beta_K$–$\alpha_J$ loop (KJ basic patch), and the other (K1488, R1494, and K1592) is located in $\alpha_K$ and $\alpha_N$ (KN basic patch). The K/R residues involved in the interaction with DNA are shown in blue. The other three panels show the surface electrostatic potential of p300$_{BRPH}$ for each complex structure, with surfaces charged positively in blue or negatively in red. Other panels (#5–#7): the surface electrostatic potential of p300$_{BRPH}$ for each complex structure. Positively charged surfaces are colored in blue and negatively charged surfaces in red. **d** Close-up views of the density and model structure of each NT in the H4acNuc complex. From left to right, the HAT catalytic center of p300 or CREB-binding protein (CBP) is shown in close proximity to H3NT (H3-I, #2), H3NT (H3-II, #10), H2ANT (#7), and H2BNT (#4) in H4acNuc. The rightmost panel showing H2BNT is another angle of Fig. 2f. Color codes of NT: blue: H3NT, yellow: H2ANT, red: H2BNT; cyan, p300 RP; magenta, p300 HAT.

(Fig. 3b). In p300$_{H2B}$·H4acNuc and p300$_{H2A}$·H4acNuc, bromodomain binding to H4NTac interacted with the minor groove at the superhelical location (SHL) + 2. In p300$_{H3-I}$·H4acNuc and p300$_{H3-II}$·H4acNuc, it interacted with the minor groove at SHL + 1. Importantly, in all structures except p300$_{H2B}$·H4acNuc, HAT of p300$_{BRPH}$ interacted with DNA through one or two basic patches (Fig. 3c). Of these, one basic patch[28] (KJ basic patch; Supplementary Fig. 7) always interacted with the nucleosomal DNA. Along with that,

another basic patch (KN basic patch) interacted with the linker DNA in p300$_{H3-I}$·H4acNuc, suggesting an acetylation mechanism for H3NT that is less dependent on pre-acetylation of H4NT. This interaction mechanism explains why H3NT closest to the linker DNA is more likely to be acetylated indiscriminately.

In p300$_{H3-I}$·H4acNuc, the density of H3NT toward HAT could be modeled for the C-terminal residues after G33 (Fig. 3d, leftmost panel). Since the distance between the substrate-binding pocket of HAT and

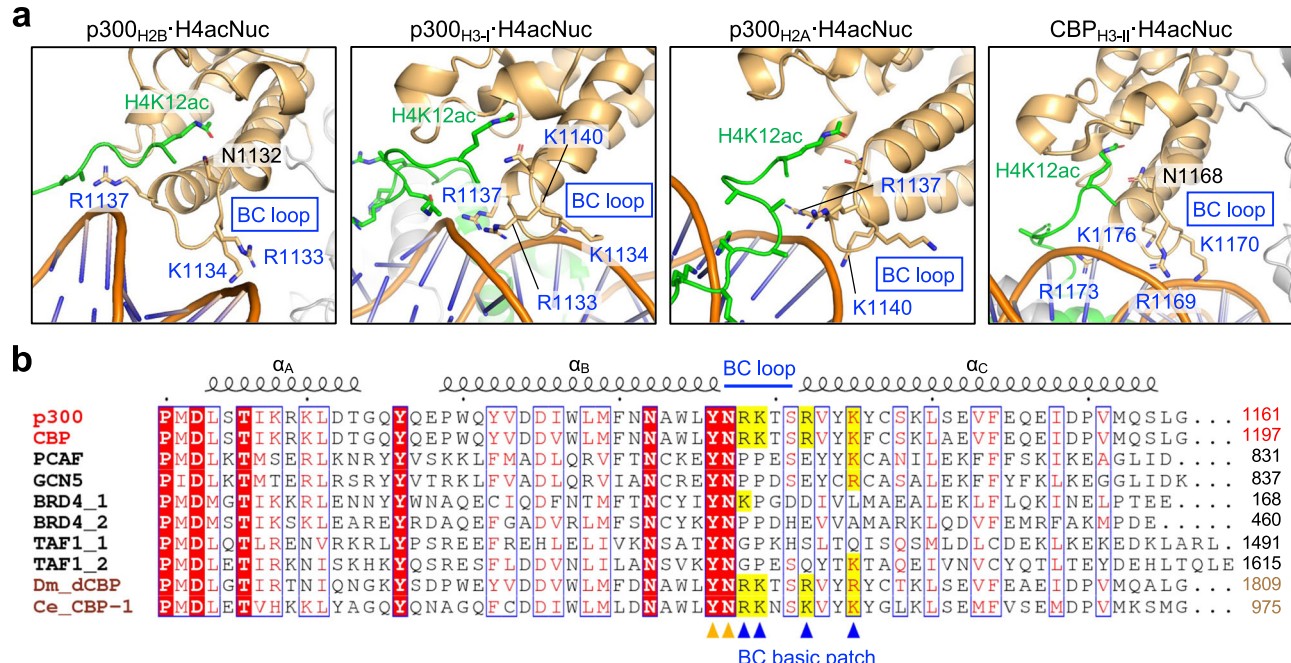

**Fig. 4 | Rotational positions of p300_BRPH/CREB-binding protein (CBP)_BRPH on the nucleosome fixed by a bromodomain loop. a** Basic patch interacting with DNA at the p300/CBP bromodomain. Close-up views (#1, #5, #7, and #10 in Supplementary Fig. 9) of the bromodomain interacting with the H4K12acK16ac region of H4-di-acetylated nucleosome (H4acNuc) in complex with p300_BRPH are shown. This basic patch (R1133, K1134, R1137, and K1140 of p300; R1169, K1170, R1173, and K1176 of CBP) is located around the BC loop of the p300/CBP bromodomain (BC basic patch; indicated by blue arrowheads at the bottom). The K/R residues involved in the interaction with DNA are shown in blue. In all structures, multiple R/K residues are involved in the interaction with DNA. Color code: light orange, the p300/CBP bromodomain; green, K12/K16-acetylated H4. **b** Sequence alignment around the BC loop of the p300/CBP bromodomain. The positions of the BC loop and three α-helices composing the bromodomain are shown on the top in blue and black, respectively. Protein names of representative human bromodomains are shown in black on the left. The protein names in brown are *D. melanogaster* Nejire/dCBP and *C. elegans* CBP-1. Residue numbers on the C-terminal side are shown on the right. Conserved or similar residues are shown in red and surrounded by blue boxes. The completely conserved residues are shown in white letters on a red background. The positions of residues involved in the recognition of acetyllysine inside the bromodomain pocket (Y1131 and N1132 of human p300) are indicated by orange arrowheads at the bottom. The positively charged residues conserved in the BC basic patch are indicated by a yellow background.

G33 is ~21 Å, p300_BRPH of this complex is assumed to acetylate from the N-terminal side of H3 up to K23. In p300_H2A·H4acNuc, the density of H2ANT toward HAT could be modeled for the C-terminal residues after K13 (Fig. 3d, second panel from the right). The distance between the substrate-binding pocket and K13 (~30 Å) suggests that p300_BRPH acetylates only K5 of H2A.

By preparing CBP_BRPHZT (residues 1084–1873) protein corresponding to the same construct as p300_BRPHZT, we also determined three complex structures (Supplementary Figs. 1f, 8, 9, Supplementary Table 3) in which HAT of CBP_BRPH is oriented toward H2BNT (CBP_H2B·H4acNuc) or H3NT (CBP_H3-I·H4acNuc and CBP_H3-II·H4acNuc). The overall conformation of CBP_BRPH·H4acNuc complexes was almost identical to that of p300_BRPH·H4acNuc, and the H4acNuc-binding modes of bromodomain and HAT were also almost identical to those of the corresponding p300_BRPH·H4acNuc structures (Supplementary Fig. 10). Whereas the inter-molecular arrangement of p300_H3-II·H4acNuc structure was determined only at 6.9 Å, the CBP_H3-II·H4acNuc structure was determined at 4.2 Å, so the domain arrangement of CBP_BRPH and the structure of the H4acNuc-interactive region in its bromodomain could be determined. HAT of CBP_BRPH in CBP_H3-II·H4acNuc was located closer to H3NT than in CBP_H3-I·H4acNuc. The density of H3NT toward HAT could be modeled for the C-terminal residues after K36 (Fig. 3d, second panel from the left). The distance between the substrate-binding pocket and K36 (~30 Å) suggests that CBP_BRPH of this complex acetylates K27 in addition to lysine residues up to K23. Collectively, these results suggest that not only p300 but also CBP can acetylate multiple non-H4 histone NTs by rotating themselves on the nucleosome with their bromodomain bound to the acetylated H4NT as the axis.

Cryo-EM maps of the histone tails with HAT of p300_BRPH or CBP_BRPH in close proximity in each structure are superimposed on the corresponding structural model (Supplementary Fig. 11). For each of these complexes, a structural comparison with the recently reported p300·nucleosome complex structure[28] is shown in Supplementary Fig. 12. In the most resolved structure by Hatazawa et al.[28] of catalytically inactive p300(BRPH_ΔAIL Z) complexed with the NCP reconstituted with a 145-bp Widom 601 sequence (complex I; PDB ID: 7W9V), its HAT and bromodomain contact nucleosomal DNA at SHLs 2 and 3. p300 in their structure does not read/write Kac on the nucleosome because bromodomain is not in close proximity to any histone NTs, although the catalytic site of HAT is near one of the H4NTs. Indeed, the position of p300 on the nucleosome in their structure is quite different from any of our structures in which p300 presumably reads/writes Kac on the nucleosome.

**Bromodomain-dependent rotation of p300/CBP**

Next, we examined why p300_BRPH or CBP_BRPH structures can direct HAT to such a variety of histone NTs when bound to H4acNuc. In all the structures obtained (five structures of p300_BRPH·H4acNuc and three structures of CBP_BRPH·H4acNuc), their bromodomain bound to H4K12ac of H4acNuc at the inside of its pocket and to the minor groove of the nucleosomal double-stranded DNA at the outside of the pocket (Fig. 4a). The position of bromodomain interacting with the minor groove varied from complex to complex, but in all structures, the third basic patch present around the BC loop (BC basic patch; Fig. 4a, Supplementary Fig. 7) interacted electrostatically with the phosphate groups of the double-stranded DNA backbone. The electrostatic interaction both stabilizes binding to H4NTac via the inside of

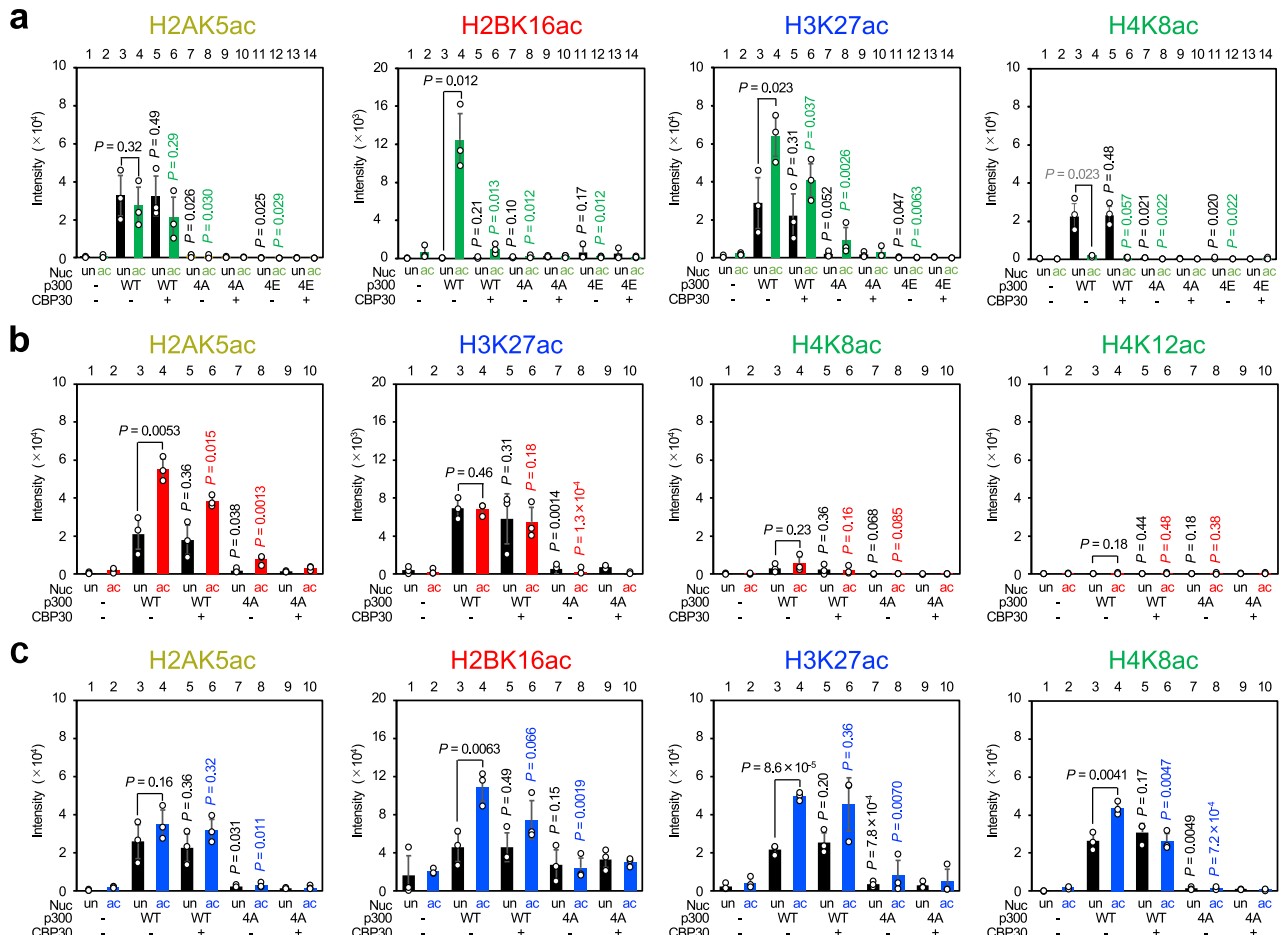

**Fig. 5 | Propagation of intranucleosomal histone acetylation by p300$_{BRPHZT}$.** **a** In vitro acetyltransferase activity of p300$_{BRPHZT}$ toward the H4-di-acetylated nucleosome. The histone and its residue at which acetylation was detected by immunoblotting are shown above each panel. Nucleosome (Nuc): un (black), unmodified; ac (green), H4K12/K16-acetylated. p300$_{BRPHZT}$ (p300): WT, wild-type; 4A, with mutations of R1133A, K1134A, R1137A, and K1140A; 4E, with mutations of R1133E, K1134E, R1137E, and K1140E. CBP30: −, none; +, 10 μM. CBP30 is an inhibitor that prevents the bromodomain pocket of p300 from binding to the acetylated histone N-terminal tails. The y-axis indicates the immunoblotting signal intensity at 1 min after the reaction. Data are mean ± SD from three independent experiments. *P* value was calculated by a two-sample one-sided Welch's *t*-test. The alternative hypothesis is as follows: lane 4, increase vs. lane 3; lanes 5, 7, and 11, decrease vs. lane 3; lanes 6, 8, and 12, decrease vs. lane 4. *P* value shown in gray is not a significant increase. **b** In vitro acetyltransferase activity of p300$_{BRPHZT}$ toward the H2B-tetra-acetylated nucleosome. Columns marked ac (in red) indicate the H2BK12/K15/K20/K23-acetylated nucleosome. Data are mean ± SD from three independent experiments. *P* value was calculated by a two-sample one-sided Welch's *t* test. The alternative hypothesis is as follows: lane 4, increase vs. lane 3; lanes 5 and 7, decrease vs. lane 3; lanes 6 and 8, decrease vs. lane 4. Other indications are the same as in **a**. **c** In vitro acetyltransferase activity of p300$_{BRPHZT}$ toward the H3-di-acetylated nucleosome. Columns marked ac (blue) indicate the H3K14/K18-acetylated nucleosome. Data are mean ± SD from three independent experiments. *P* value was calculated by a two-sample one-sided Welch's *t* test. The alternative hypothesis is the same as in **b**.

the bromodomain pocket and alters the relative positioning of HAT on the nucleosome, depending on where it occurs. Interestingly, neither p300$_{BRPH}$ nor CBP$_{BRPH}$ recognized any DNA sequences. Consequently, having DNA sequence–independent multivalent modes of binding to the nucleosome presumably allows HAT of p300/CBP to successively acetylate any of the non-H4 histone NTs within the H4NT-acetylated nucleosome.

Sequence alignment of bromodomains indicates that R/K residues composing the BC basic patch (i.e., RKxxRxxK in p300/CBP, where x indicates an unrelated residue) are conserved only in p300 and CBP among all 61 human bromodomains (Fig. 4b, Supplementary Fig. 13). Importantly, these four R/K residues are conserved among metazoan p300 homologs, including *D. melanogaster* Nejire/dCBP[36] and *C. elegans* CBP-1[37].

**The BC basic patch is critical for read/write**

To investigate the function of the BC basic patch in the read/write mechanism, we prepared mutant proteins of p300$_{BRP}$ or p300$_{BRPHZT}$ in which all four positively charged residues were mutated with alanine

(4A) or glutamic acid (4E) residues, respectively. First, concerning the read mechanism, we measured the half-saturation concentration ($K_{1/2}$) of p300$_{BRP}$ for nucleosome binding by microscale thermophoresis with and without H4NTac (i.e., K12ac/K16ac), p300 4A mutation, and CBP30 (Supplementary Fig. 14; Supplementary Table 4). Wild-type p300$_{BRP}$ bound to the unmodified nucleosome relatively strongly with a $K_{1/2}$ of 2.2 ± 0.51 nM. As expected, this binding was enhanced sixfold by the presence of H4NTac ($K_{1/2}$ = 0.35 ± 0.10 nM), which was largely canceled by prior incubation with CBP30 ($K_{1/2}$ = 1.2 ± 0.20 nM). By contrast, 4A weakened the affinity for H4acNuc by fivefold compared to the wild-type ($K_{1/2}$ = 1.8 ± 0.22 vs. 0.35 ± 0.10 nM). Pre-incubation with CBP30 further weakened this affinity by twofold ($K_{1/2}$ = 3.5 ± 1.6 nM). These results reinforce our structures, in which the binding of p300$_{BRPH}$ to H4acNuc is mediated both inside and outside the bromodomain pocket.

We next examined the effects of the two mutations of p300$_{BRPHZT}$ on the H4NTac-dependent p300 read/write mechanism (Fig. 5a, Supplementary Fig. 15). Both mutations dramatically reduced the H4NTac-dependent acetylation of H2BNT at all residues examined. They also

reduced the H4NTac-dependent H3NTac found at K14, K23, and K27. This reduction was more prominent for 4E mutation, in which positively charged residues became negatively charged, than for 4A, in which residues became uncharged. H2BNTac was markedly reduced by CBP30 alone, and H3NTac was also almost completely suppressed by combining it with either mutation. These results suggest that the multivalent binding of p300$_{BRPHZT}$ to H4acNuc is critical for Kac propagation to H2BNT and H3NT. Interestingly, both mutations also markedly reduced the H4NTac-independent acetylation by p300$_{BRPHZT}$ (i.e., H2AK5ac, H3K18ac, and H4K5ac). Collectively, the BC basic patch is suggested to help p300 bind to and acetylate the nucleosome not only in an H4NTac-dependent manner but also independently of histone NTac.

### Directionality of histone NT acetylation

When acetylation was present in H4NT, p300$_{BRPHZT}$ facilitated multisite lysine acetylation of the NTs in H2B and H3 within the nucleosome in a bromodomain-dependent manner. So, when acetylation is present in the non-H4 histone NTs, would p300 facilitate the acetylation of other histone NTs? To this end, we reconstituted the nucleosomes containing either H2B acetylated at K12/K15/K20/K23 (H2BacNuc) or H3 acetylated at K14/K18 (H3acNuc) and examined whether p300$_{BRPHZT}$ acetylates them (Supplementary Fig. 16). H2BacNuc did not significantly facilitate p300$_{BRPHZT}$-catalyzed NTac at the one-minute timepoint, except H2AK5ac (Fig. 5b, Supplementary Fig. 17). On the other hand, H3acNuc significantly facilitated p300$_{BRPHZT}$-catalyzed H2BNTac and H4NTac at the one-minute timepoint (Fig. 5c, Supplementary Figs. 18, 19a). In all cases, including no pre-acetylation, p300$_{BRPHZT}$ did not acetylate H4K16 at all and hardly acetylated H4K12. Collectively, the directionality in which p300$_{BRPHZT}$ reads/writes Kac between NTs falls into two types: (1) bidirectional between H4NT and H3NT and (2) unidirectional from H4NT to H2BNT or from H3NT to H2BNT. This suggests that p300 bromodomain binds to H4NTac and H3NTac as reported[15], but cannot or very weakly binds to H2BNTac.

Interestingly, when the nucleosome was pre-acetylated at H3K14/K18, p300$_{BRPHZT}$ significantly propagated acetylation to K23/K27 (Fig. 5c, Supplementary Fig. 18). Since it is structurally difficult for p300/CBP to simultaneously read/write K14/K18 and K23/K27 in one H3NT, this result suggests that p300$_{BRPHZT}$ propagates lysine acetylation across the two H3NTs. Indeed, our model suggested that the H3NT pair in the nucleosome is close enough in proximity that p300$_{BRPH}$ can read/write H3NT(ac) on both (Supplementary Fig. 19b).

### H2BNTac destabilizes the nucleosome

When the multisite H2BNTac is triggered by H3NTac or H4NTac, are there any proteins recruited subsequently? The multisite H4NTac, such as H4K5ac/K8ac, is a scaffold to which the bromodomains of proteins in the bromodomain and extra-terminal (BET) family preferentially bind[38,39]. Since the sequence between K5ac and K8ac of H4NT is similar to that between K12ac and K15ac of H2BNT[40], the BET bromodomains may bind to the di-acetylated H2BNT. To this end, we examined dissociation constants ($K_D$) between p300 or BET bromodomains and several di-acetylated H2BNT or H4NT peptides by isothermal titration calorimetry (Supplementary Fig. 20; Supplementary Table 5). BRD4$_{BD1}$, the N-terminal bromodomain of BET family protein BRD4, bound very weakly or poorly to the di-acetylated H2B peptides tested, with a minimal $K_D$ of $430 \pm 5.0\,\mu M$ (i.e., for K12ac/K15ac). This affinity was 20-fold weaker than the $K_D$ between BRD4$_{BD1}$ and the H4K12ac/K16ac peptide ($22 \pm 7.0\,\mu M$). Hence, it is unlikely that multisite H2BNTac could recruit BET proteins as the multisite H4NTac does. Additionally, bromodomain-containing p300$_{BRP}$ also bound very weakly or poorly to the di-acetylated H2B peptides (K20ac/K23ac), showing a minimal $K_D$ of $200 \pm 30\,\mu M$. This affinity was also 13-fold weaker than the $K_D$ between p300$_{BRP}$ and the H4K12ac/K16ac peptide

($15 \pm 9.0\,\mu M$). Therefore, multisite H2BNTac is an unlikely scaffold for nucleosome binding either by p300 or BRD4 bromodomains. This is consistent with the fact that the multisite H2BNTac did not facilitate the p300$_{BRPHZT}$-catalyzed histone NTac, except for H2AK5ac.

Thus, multisite H2BNTac may serve as an endpoint of this signaling rather than recruiting other proteins. If so, it is natural to assume that the high correlation between H2BNTac and the enhancer activity[20,21] is not the result of transcription-coupled histone exchange, but rather its direct cause. Based on this hypothesis, we examined the effect of H2BNTac on the thermal stability of the nucleosome. Of the stepwise dissociation of the H2A-H2B dimer and the H3-H4 tetramer, the lower melting temperature ($T_m$), reflecting the dissociation of the H2A-H2B dimer, decreased 0.6 °C for H3acNuc and increased 0.1 °C for H4acNuc compared to the unmodified nucleosome (Supplementary Fig. 21), but decreased 1.9 °C for H2BacNuc (73.0 vs. 74.9 °C; Fig. 6a). The higher $T_m$, reflecting the dissociation of the H3-H4 tetramer, was similar to that in the unmodified nucleosome at any acetylation (82.9–83.1 °C vs. 83.3 °C for the unmodified nucleosome). These results suggest that multisite H2BNTac selectively promotes H2A-H2B dissociation from the nucleosome.

## Discussion

Since the study of Allfrey et al.[5], histone acetylation has become recognized as an important PTM regulating eukaryotic gene transcription. Among histone acetyltransferases, p300/CBP rapidly activates the transcription of specific enhancer-regulated genes by binding to various DNA sequence–binding transcription factors (TFs) and through its own acetyltransferase activity[25], which is important for the regulation of diverse cellular responses and diseases. The present study revealed how p300/CBP recognizes H4NTac and propagates lysine acetylation to non-H4 histone NTs within the nucleosome. That is, p300/CBP reads H4NTac at the bromodomain pocket, rotates in multiple directions, and rapidly writes NTac to non-H4 histones within the nucleosome independently of the DNA sequence. To our knowledge, this is the first structural evidence showing how a particular PTM in the nucleosome is read/written by the enzyme to self-propagate. In contrast to the mechanisms by which histone methylation at H3K9 and H3K27 involved in transcriptional repression spreads to neighboring nucleosomes[41], histone acetylation involved in transcriptional activation spreads within a single nucleosome. The read/write role of p300/CBP derived from our data is twofold: (1) replication of NTac within the H3-H4 tetramer and (2) transcription of NTac from the H3-H4 tetramer to the H2B-H2A dimer.

The first role for p300/CBP was derived from the symmetry in the flow of acetylation information between H3NT and H4NT (Supplementary Fig. 19a). The distance between the H4NT pair within the nucleosome was too far for p300/CBP to read/write Kac directly between them. However, when bound to one of a pair of H4NTs, p300/CBP was at the right proximity to acetylate the closer NT for each of the pairs of H3NT, H2BNT, and H2ANT. Our data also suggest that p300$_{BRPH}$ can propagate Kac between the H3NT pair (Supplementary Figs. 18, 19b). Therefore, p300/CBP may read/write the multisite NTac 1) from one of the H4NT pair to the proximal H3NT, 2) between the H3NT pair, and 3) from the distal H3NT to the other H4NT. p300/CBP would be a maintenance acetyltransferase that would ensure self-perpetuation of Kac in the H3-H4 tetramer.

To ensure the self-perpetuation of Kac in the H3-H4 tetramer, it would be important for p300/CBP to preferentially bind to nucleosomes containing H4K12ac/K16ac ($K_D = 25\,\mu M$), H4K8ac/K12ac ($K_D = 90\,\mu M$), or H3K14ac/K18ac ($K_D = 104\,\mu M$) via bromodomain[15]. Indeed, acetylation of these five residues (i.e., H4K8ac, H4K12ac, H4K16ac, H3K14ac, and H3K18ac) is stably maintained during mitosis in mammalian cells[29], suggesting that they are already acetylated in the early $G_1$ phase just after mitosis. For H4NTac in particular, H4K8ac is a mitotic bookmark for GAGA pioneer factor to activate its target genes

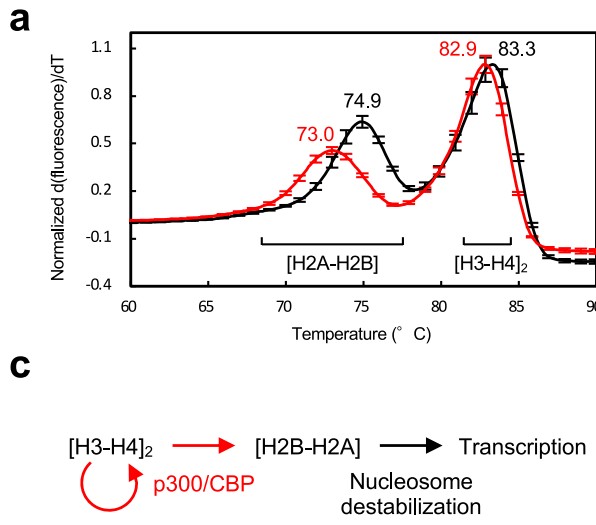

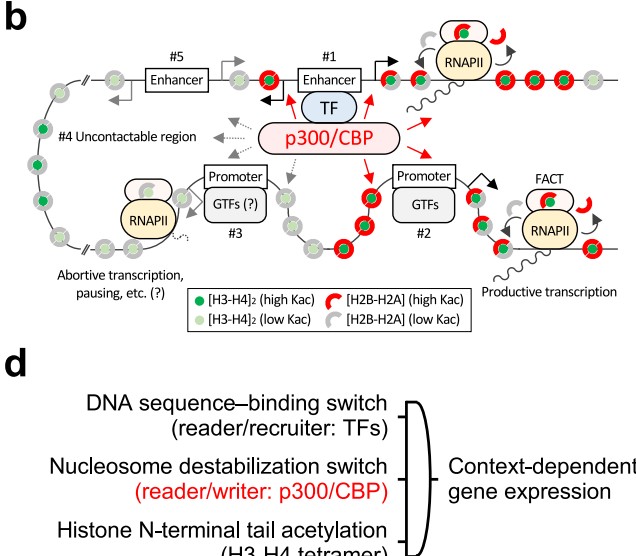

**Fig. 6 | Model for regulation of chromatin transcription by p300/CREB-binding protein (CBP). a** Thermal stability assay of the H2BNT-acetylated nucleosome. Mean values of thermal denaturation curves from 60.0 °C to 90.0 °C for derivative fluorescence intensity are plotted for the unmodified nucleosome (black line) and the H2BK12/K15/K20/K23-acetylated nucleosome (red line). Melting temperature ($T_m$) is shown at each peak. The temperature at which the H2A-H2B dimer or the H3-H4 tetramer dissociates from the nucleosome is shown at the bottom. Data are mean ± SD from three independent experiments. **b** Schematic model of p300/CBP-driven chromatin transcription regulation. p300/CBP, which is recruited by its partner transcription factor (TF) binding to an enhancer, acetylates chromatin in the proximity (#1) and in contactable regions (#2) where the H3-H4 tetramer is pre-acetylated more than usual. The acetylation of H2B that p300/CBP transcribes from acetylation of H3/H4 promotes dissociation of the H2B-H2A dimer from the nucleosome and determines the chromatin regions where the RNA polymerase II (RNAPII)·FAcilitates Chromatin Transcription (FACT) complex readily exchanges the H2B-H2A dimer for productive transcription. This reaction is less likely to occur in hypoacetylated regions (#3), uncontactable regions (#4), or partner TF-unbound regions (#5). GTFs, general transcription factors; Kac, lysine acetylation. **c** Epicentral model of histone acetylation signaling. Arrows indicate the flow of information, with acetylation information in red. **d** Hypothetical logic of context-dependent gene expression in metazoans. The symbol in the center indicates a triple-input AND logic gate.

in fly[42]. H4K12ac, which is recognized by the bromodomain pocket of p300$_{BRPH}$, is already acetylated in the cytoplasm[43–45] and maintained during mitosis in living cells[46]. Also, H4K16ac is intergenerationally maintained in fly and mammals, making it instructive for future gene activation[47]. However, our data and those of another group[17] show that p300 can hardly acetylate nucleosomal H4 at K16 and not much at K12. Given the distance between the H4NT pair, these data suggest that H4K8ac/K12ac self-perpetuates by p300/CBP with the help of H3K14ac/K18ac and cytoplasmic acetylation of H4K12. Self-perpetuation of H4K16ac presumably requires the MYST (MOZ, YBF2, SAS2, and Tip60)-type histone acetyltransferases, such as MOF/KAT8[48,49]. For H3NTac, the distance between the H3NT pair suggests that H3K14ac/K18ac may self-perpetuate by p300/CBP alone (Supplementary Fig. 19b). Such p300/CBP-mediated self-perpetuation of Kac in the H3-H4 tetramer would contribute to both the inheritance of Kac information to daughter cells and the p300/CBP-driven rapid transcriptional activation of specific genes immediately after mitosis[50].

How is acetylation that is propagated within the H3-H4 tetramer inherited across the cell cycle? Individual histone octamers segregate conservatively[51] during DNA replication and are redeposited at the same position in one of the two daughter DNA molecules[52]. Intriguingly, the H3-H4 tetramer partially splits into H3-H4 dimers[53] at transcriptionally active genes and enhancers[54,55], possibly with the help of the histone chaperone ASF1[56–58]. Therefore, the acetylation information of the parental histones may be inherited by the H3-H4 tetramer, consisting of the parental H3-H4 dimer and a new H3-H4 dimer, in both transcriptionally active daughter chromatin regions. Then, p300/CBP may restore the diluted acetylation information derived from the parental H3-H4 dimer (Supplementary Fig. 19a), some of which may be inherited by both daughter chromatin by escaping global histone deacetylation during mitotic entry[59]. Mechanistically, p300/CBP-driven NTac and/or H3K56ac[60] may promote ASF1-dependent H3-H4 tetramer splitting and its semi-conservative redeposition.

The second role for p300/CBP was derived from the asymmetry in the flow of acetylation information from H4NT or H3NT to H2BNT. p300/CBP interacts with at least 400 proteins, including various DNA-binding TFs[61]. Presumably, p300/CBP binds to the partner TFs bound to specific DNA sequences via its intramolecular domain(s) such as TAZ1[62,63] and TAZ2[8,64–67], while simultaneously interacting via bromodomain[14,15] and ZZ[30] with nucleosomes not only proximal to the enhancers but also distal ones by chromatin looping (Fig. 6b). Then, p300/CBP acetylates H2B (e.g., at active enhancers, their target promoters, and gene body regions) of those nucleosomes when H3NTac or H4NTac is present. This would promote dissociation of the H2A-H2B dimer from the nucleosome in these regions, at which RNAPII enters DNA more readily and/or transcribes it more productively[25].

Importantly, TF-dependent histone acetylation by p300 facilitates the transfer of the H2A-H2B dimers from the nucleosomes to the histone chaperone NAP-1, possibly via its interaction with acetylated H2A-H2B[68]. In this light and given other findings on the behavior of H2B[69–71], it makes sense that p300-catalyzed H2BNTac is the genuine signature of active enhancers and their target promoters[20,21]. In addition, RNAPII complexed with the histone chaperone FACT flips the histone octamer and exchanges one H2A-H2B dimer during traverse across the nucleosome[72]. Thus, H2BNTac information written by p300/CBP should be erased by subsequent successive transcription, and the H2BNTac information that guides transcription should be rapidly and repeatedly updated. H3NTac-activated histone exchange from H2A to H2A.Z on chromatin[73] may further facilitate nucleosomal destabilization and RNAPII recruitment at promoters through p300-driven multisite acetylation of H2A.Z[16]. It should be noted, however, that our study does not make clear how the thermal instability of the nucleosome caused by the p300-catalyzed multisite H2BNTac, per se, affects histone exchange under physiological conditions, nor whether p300 acetylates H2BNT most in the cell by the present structural

mechanism. These points are limitations of this study and need further biochemical and cell biological validation.

The asymmetry in the flow of acetylation information would originate from a KK sequence unique to H2BNT (Supplementary Fig. 16b). Crystal structures[14,15] suggest that a residue having a long side chain such as lysine (K) just before or after Kac prevents p300 bromodomain binding, and indeed this sequence is absent in H3NT and H4NT. H2BNT is less conserved than H3NT and H4NT, but multiple KK sequences are conserved in H2B from yeast to human, a feature not found in other histones[21]. This KK sequence may be a mechanism devoted to nucleosome destabilization via dense multisite lysine acetylation, preventing unnecessary signaling via the Kac readers. Also, acetylation of H2BNT was structurally the least dependent on the DNA-binding activity of p300 HAT among the histone NTs (Fig. 3b, c). This suggests that H2BNTac is the least indiscriminately acetylated, strictly p300 reader activity-regulated read/write switch with the best signal-to-noise ratio.

The specificity of chromatin acetylation has been attempted to be explained primarily by one of the following two models: (1) a TF binding to a specific DNA sequence specifies the chromatin sites where its partner acetyltransferase (complex) acetylates, or (2) an acetyltransferase (complex) binding to histone NTac specifies the chromatin sites to acetylate independently of DNA sequence. The current situation is more supportive for the former because the epigenomic locations of p300 largely overlap with those of its partner TFs independently of its acetyltransferase activity[25], and the acetyltransferase activity of p300 directly depends on activation of TF ligands[15]. On the other hand, support for the latter is that p300 bromodomain is critical in modulating its enzymatic activity and its association with chromatin[30] and also that inhibition of the p300/CBP bromodomain pockets decreases H3K27ac and transcription of enhancer-proximal genes[74]. Based on the present data, we support a model that integrates both (Fig. 6b). That is, the global specificity of chromatin acetylation would be determined by the proximity or contactable range of the chromatin to p300/CBP, which is recruited by its partner TF bound to a specific DNA sequence in the genomic DNA. Subsequently, the local specificity of chromatin acetylation would be determined by whether the H3-H4 tetramer is pre-acetylated (even slightly) more than usual when p300/CBP interacts with each nucleosome of that chromatin.

Finally, we propose an epi-central model of p300/CBP-catalyzed histone acetylation signaling (Fig. 6c). Here, acetylations of the H3-H4 tetramer and the H2A-H2B dimer play distinct roles. Just as DNA is the genetic storage, p300/CBP replicates histone acetylation within the H3-H4 tetramer, which self-perpetuates as epigenetic storage. Also, just as RNA is the genetic processor, p300/CBP transcribes histone acetylation from the H3-H4 tetramer to the H2B-H2A dimers in a strictly regulated manner, which then self-sacrifices to express specific genes as the epigenetic processor. In other words, the essence of gene regulation unique to eukaryotes would be that the nucleosome has one $[H3-H4]_2$ tetramer in the inner core that inherits epigenetic information and two $[H2A-H2B]$ dimers in the outer shell that express that information (Fig. 6b). Lysine acetylation in the nucleosome would have a duality of epigenetic inheritance and expression, depending on which histone it is in. Regarding epigenetic switches, a DNA sequence–binding protein that may self-perpetuate by a positive feedback mechanism is a read/recruit switch that specifies a gene to be transcribed from genomic DNA[75–77], and p300/CBP is presumably a nucleosome destabilization read/write switch for productively transcribing that gene from the nucleosomes (Fig. 6d). The logic of context-dependent gene expression in metazoans would be a triple-input AND-gated circuit consisting of the DNA sequence–binding switch[75], the nucleosome destabilization switch, and histone N-terminal tail acetylation of the H3-H4 tetramer[5–7].

## Methods

### Expression and purification of p300, CBP, and BRD4

The cDNA encoding human $p300_{BRPHZT}$ (residues 1048–1836) and $CBP_{BRPHZT}$ (residues 1084–1873) were amplified by PCR and subcloned into a pFastBac HT vector (Thermo Fisher Scientific, 10584027) with a glutathione S-transferase (GST, GenBank accession number M14654)-encoding sequence inserted between an N-terminal 6×His tag and a tobacco etch virus (TEV) protease recognition site (ENLYFQG) in the vector. Site-directed mutagenesis of the $p300_{BRPHZT}$ catalytically inactive (Y1467F), 4A-substituted (R1133A, K1134A, R1137A, and K1140A), and 4E-substituted (R1133E, K1134E, R1137E, and K1140E) mutant proteins was performed by PCR, using the DpnI restriction enzyme. All introduced mutations were verified by DNA sequencing. The $p300_{BRPHZT}$, $CBP_{BRPHZT}$, and $p300_{BRPHZT}$ Y1467F, 4A-, and 4E-substituted mutant proteins were expressed in baculovirus-infected High Five insect cells (Thermo Fisher Scientific, B85502). Baculoviruses were produced using the Bac-to-Bac baculovirus expression system (Invitrogen). The baculovirus-infected cells were collected 72 hrs after transfection, and cell pellets were frozen at −80 °C until purification. Frozen High Five cells were resuspended in 20 mM Tris-HCl buffer (pH 7.2) containing 500 mM NaCl, 10% glycerol, 20 mM imidazole, 0.1% NP-40, 1.5 mM $MgCl_2$, 1 μM $ZnCl_2$, DNase I (Sigma Aldrich), and cOmplete (EDTA-free) Protease Inhibitor Cocktail (Roche). Cells were lysed by sonication and clarified by 60-min centrifugation at $30,000 \times g$ at 4 °C. The cell lysate from each sample was loaded onto a HisTrap HP column (Cytiva) and eluted using 50 mM Tris-HCl buffer (pH 8.0) containing 500 mM NaCl, 10% glycerol, and 500 mM imidazole. After buffer exchange using a HiTrap Desalting column (Cytiva), the N-terminal 6×His-GST tag was cleaved by incubation with TEV protease at 4 °C overnight. The cleaved protein was then reapplied to a HisTrap HP column, and the flow-through fraction was collected. The collected fractions were purified by size-exclusion column chromatography, using a HiLoad Superdex 200 26/60 (Cytiva) equilibrated with 20 mM HEPES-NaOH buffer (pH 7.2) containing 250 mM NaCl, 1 mM Tris (2-carboxyethyl) phosphine (TCEP), and 5 μM $ZnCl_2$. Purified protein was concentrated using an Amicon Ultra-15 centrifugal filter unit (Merck Millipore, 50 kDa MWCO) and flash-frozen in liquid nitrogen. $p300_{BRPHZT}$ and $p300_{BRPHZT}$ Y1467F were electrophoresed in a 10–20% sodium dodecyl sulfate (SDS) polyacrylamide gel (DRC, NXV-396HP20) and transferred onto a nitrocellulose membrane (BIO-RAD, 1620112) at 20 V for 10 min by the semi-dry method. The membrane was blocked with Bullet Blocking One for Western Blotting (Nacalai Tesque, 13779-01) for 10–30 min at 25 °C. Membranes were incubated for 20–40 min at 25 °C with Bullet ImmunoReaction Buffer (Nacalai Tesque, 18439-85). Autoacetylation of $p300_{BRPHZT}$ and $p300_{BRPHZT}$ Y1467F were analyzed by immunoblotting using anti-p300K1499ac antibody (Cell Signaling, 4771, dilution rate: 1/1000) and horseradish peroxidase (HRP)-conjugated anti-rabbit IgG antibody (Cytiva, NA934, 1/10000). Uncropped gels of SDS polyacrylamide gel electrophoresis (SDS-PAGE) and immunoblotting are shown in Supplementary Fig. 22.

The cDNA encoding human $p300_{BRP}$ (residues 1048–1282) and the N-terminal bromodomain of human BRD4 (residues 44–168; $BRD4_{BD1}$) were amplified by PCR and subcloned into a pET28a(+)-derived plasmid containing the N-terminal 6×His tag, the GST sequence, and a TEV protease recognition site (EHLYFQG). Site-directed mutagenesis of $p300_{BRP}$ 4A-substituted (R1133A, K1134A, R1137A, and K1140A) mutant was performed by PCR using the DpnI restriction enzyme. All introduced mutations were verified by DNA sequencing. $p300_{BRP}$ (i.e., the wild-type and the 4A-substituted mutant) and $BRD4_{BD1}$ were expressed in LB broth of *E. coli* BL21 (DE3) cells at 37 °C until the $OD_{600}$ reached 0.8. The temperature was then shifted to 18 °C, and isopropyl-β-D-thiogalactopyranoside (IPTG) was added to a final concentration of 300 μM to induce protein expression. The cultures were incubated for an additional 20 hrs and cells were collected by 5-min centrifugation at $7000 \times g$ at 4 °C. Cell

pellets were resuspended in 50 mM Tris-HCl buffer (pH 8.0) containing 500 mM NaCl, 10% glycerol, and 20 mM imidazole. For p300$_{BRP}$, the buffer also contained 10 µM ZnCl$_2$. Cell lysates prepared by sonication and 30-min centrifugation at 30,000 × $g$ at 4 °C were purified on a HisTrap HP column (Cytiva). The N-terminal 6×His-GST tag was cleaved by incubation with TEV protease at 4 °C overnight. The cleaved protein was then applied to a GSTrap HP column (Cytiva), and the flow-through fraction was collected. The eluted fractions were loaded on a HiLoad Superdex 200 16/60 column (Cytiva) equilibrated with 10 mM HEPES-NaOH buffer (pH 7.4) containing 150 mM NaCl.

## Cell-free synthesis of residue-specific acetylated histones

Recombinant human histone proteins containing residue-specific Kac(s) (K12/K15/K20/K23-acetylated H2B, K14/K18-acetylated H3, K14/K18/K23/K27-acetylated H3, K8/K12-acetylated H4, and K12/K16-acetylated H4) were synthesized by genetic code reprogramming in a coupled transcription–translation cell-free system. The cDNA containing human histone H2B type 1-J, H3.1, or H4 ORF, with the codon(s) of the specified residue(s) replaced with the TAG triplet(s) and a terminal TAA stop codon, were subcloned into a pCR2.1-TOPO plasmid (Thermo Fisher Scientific, K450002). Each histone protein was designed to have an N-terminal histidine-rich affinity tag (N11; MKDHLIHNHHKHEHAHA) followed by a TEV protease recognition site (EHLYFQ). The H2B expression construct contained a glycine residue at the end of the TEV recognition sequence (i.e., EHLYFQG) which remains after cleavage by TEV protease. All other histones have no additional N-terminal sequence after cleavage by TEV protease. These plasmids were used as the templates in a coupled transcription–translation cell-free system with a 9-ml reaction solution dialyzed against a 90-ml external feeding solution. The 9-ml reaction solution contained 0.37 volume of the low-molecular-weight creatine phosphate tyrosine (LMCPY) mixture [160 mM HEPES-KOH buffer (pH 7.5), containing 4.1 mM L-tyrosine, 3.5 mM ATP, 2.4 mM each of GTP, CTP, UTP, 0.22 mM folic acid, 1.8 mM cAMP, 74 mM ammonium acetate, 210 mM creatine phosphate, 5 mM DTT, 530 mM potassium L-glutamate, and 11% PEG8000], 0.075 volume of the 19-amino acid mixture [20 mM each of the amino acids other than L-tyrosine], 0.01 volume of 17.5 mg/ml tRNA, 0.26 volume of *E. coli* S30 extract from RFzero strains[78,79], 14 mM Mg(OAc)$_2$, 10 mM acetyllysine, 5 mM nicotinamide, 0.05% NaN$_3$, 250 µg/ml creatine kinase, 8 µM pyrrolysine-specific tRNA (tRNA$^{Pyl}$)[79], 8 µM pyrrolysyl-tRNA synthetase (PylRS) mutant KacRS_6mt[79], 67 µg/ml T7 RNA polymerase, and 16–64 µg/ml template plasmid. The 90-ml external feeding solution contained 0.03 volume of 10×S30 buffer [100 mM Tris-acetate buffer (pH 8.2), containing 600 mM potassium acetate, 160 mM Mg(OAc)$_2$, and 10 mM DTT], 0.37 volume of the LMCPY mixture, 0.075 volume of the 19-amino acid mixture, 14 mM Mg(OAc)$_2$, 20 mM acetyllysine, 10 mM nicotinamide, and 0.05% NaN$_3$. Cell-free synthesis in the dialysis mode was performed at 37 °C for 6–18 hr, using 30 cm$^2$ of dialysis membrane per ml reaction solution. The reaction mixture was centrifuged for 30 min at 30,000 × $g$ at 4 °C. Protein precipitates were used for histone octamer preparation.

## Expression of unmodified histones

Unmodified human histone proteins were recombinantly expressed in *E. coli*. The cDNA containing human histone H2A type 1-B/E, H2B type 1-J, H3.1, and H4 were amplified by PCR and subcloned into a pET15b-derived plasmid containing an N-terminal 6×His tag (MGSSHHHHHHSSG) in which the thrombin cleavage site of pET15b (Merck Millipore, 69661) was replaced by a TEV protease recognition site (EHLYFQ). The H2B expression construct contained a glycine residue at the end of the TEV recognition sequence (i.e., EHLYFQG) which remains after cleavage by TEV protease. All other histones have no additional N-terminal sequence after cleavage by TEV protease. The unmodified histones H2A type 1-B/E, H2B type 1-J, and H3.1 were

expressed in LB broth of *E. coli* BL21 (DE3) cells at 37 °C. The unmodified histone H4 was expressed in LB broth of *E. coli* JM109 (DE3) cells at 37 °C. When the OD$_{600}$ reached 0.8, IPTG was added to a final concentration of 1 mM. The cultures were incubated for an additional 16–20 hrs and cells were collected by 5-min centrifugation at 7000 × $g$ at 4 °C. Cell pellets were resuspended in 50 mM Tris-HCl buffer (pH 7.5) containing 500 mM NaCl, 20 mM imidazole, 0.1% Triton X-100, and cOmplete (EDTA-free) Protease Inhibitor Cocktail (Roche). The cells were sonicated and centrifuged for 30 min at 30,000 × $g$ at 4 °C. Protein precipitates were washed three times with 50 mM Tris-HCl buffer (pH 7.5) containing 150 mM NaCl and used for histone octamer preparation.

## Histone octamer preparation

All histone proteins synthesized in a coupled transcription–translation cell-free system or expressed in *E. coli* were insoluble, as assessed by centrifugation. The protein precipitates were solubilized in 50 mM Tris-HCl buffer (pH 8.0) containing 6–7 M guanidine-hydrochloride and 10 mM DTT. The sample was centrifuged for 30 min at 30,000 × $g$ at 4 °C and the supernatant was filtered through 0.45 µm and 0.22 µm pore size filters (Merck Millipore). Each histone sample was loaded onto a HisTrap HP column (Cytiva) and eluted with an imidazole gradient from 20 mM to 500 mM under 50 mM Tris-HCl buffer (pH 8.0) containing 500 mM NaCl, 5% glycerol, and 7 M urea. The eluate was dialyzed three times against cold distilled water containing 5 mM 2-mercaptoethanol for 2 hrs at 4 °C. The N-terminal N11 or 6×His tag was cleaved by incubation with TEV protease at a molar ratio of histone:TEV protease = 5–10:1 at 4 °C overnight. The solution of cleaved histone protein was adjusted to contain 7 M urea. The sample solution was then reapplied to a HisTrap HP column (Cytiva) and the flow-through fraction was collected. The histone proteins were subjected to ion-exchange chromatography on a Mono S 10/100 GL column (Cytiva, 17516901). The eluate was dialyzed four times against cold distilled water for 3 h at 4 °C. Purified histones with or without Kac were lyophilized. The purified histones (i.e., H2A, H2B, H3, and H4) were mixed at an equimolar ratio in 10 mM Tris-HCl buffer (pH 7.6) containing 7 M guanidine-hydrochloride and 5 mM DTT and dialyzed four times against 10 mM Tris-HCl buffer (pH 7.6) containing 2 M NaCl and 5 mM DTT for 3 hrs at 4 °C. Histone octamers were purified by gel filtration chromatography using a HiLoad Superdex 200 16/60 column (Cytiva) under 10 mM Tris-HCl buffer (pH 7.6) containing 2 M NaCl and 5 mM DTT and concentrated in Amicon Ultra-15 centrifugal filter units (Merck Millipore, 30 kDa MWCO).

## Nucleosome reconstitution

Reconstituted nucleosomes consisted of the histone octamer with the designed Kac(s) and the palindromic 180-bp DNA that consisted of the 146-bp human α-satellite DNA and 17-bp linker DNA (5′-ATC CGT CCG TTA CCG CC-3′) linked at both ends[80] or the palindromic 146-bp human α-satellite DNA without the linker DNA[2]. For the preparation of the palindromic 180-bp DNA, pWMD01–105 × 18 plasmid containing an 18 tandem 105-bp half-nucleosome DNA unit[80] was digested by EcoRV (Takara, 1042A) at 15–30 units/nmol of EcoRV site at 37 °C overnight to excise each 105-bp unit DNA. Enzymatic reactions were performed at a DNA concentration of 1 mg/ml unless otherwise noted. The digested EcoRV fragments were separated from linearized plasmids by PEG precipitation, using 9% PEG 6000 and 500 mM NaCl. DNA was purified twice by CIA (chloroform:isoamylalcohol=24:1) extraction. The 105-bp unit DNA was dephosphorylated by calf intestine alkaline phosphatase (CIAP; Takara, 2250B) at 1 unit/nmol of DNA end at 37 °C overnight. For complete dephosphorylation, CIAP was added at 0.5 units/nmol of DNA end and the reaction mixture was further incubated at 37 °C overnight. The DNA was purified twice by phenol/CIA (phenol:chloroform:isoamylalcohol = 25:24:1) extraction and precipitated by the addition of 2.5 volume of ethanol and 0.1 volume of 3 M sodium

acetate (pH 5.2). The DNA precipitates rinsed with 70% ethanol were resuspended in 10 mM Tris-HCl buffer (pH 8.0) containing 0.1 mM EDTA. The DNA was digested by EcoRI (Takara, 1040A) at 30 units/nmol of EcoRI site at 37 °C overnight to generate cohesive ends for the self-ligation. The resultant 90-bp cohesive DNA was subjected to ion-exchange chromatography on a TSKgel DEAE-5PW column (TOSOH BIOSCIENCE, 0007574) using 10 mM Tris-HCl buffer (pH 8.0) containing 0.1 mM EDTA (250–550 mM NaCl gradient). The collected DNA was purified by ethanol precipitation. The 90-bp DNA was ligated to each other by using *E. coli*. DNA ligase (Takara, 2160A) at 1 unit/μg of DNA fragment at 16 °C overnight. The resultant palindromic 180-bp nucleosome DNA was purified by ethanol precipitation.

For the preparation of the palindromic 146-bp DNA, a pUC-based plasmid containing a 24 tandem half-nucleosome human α-satellite DNA unit[2] (a gift from Dr. Timothy J. Richmond, ETH Zürich, Switzerland) was digested by EcoRV (Takara, 1042A) at 15–30 units/nmol of EcoRV site at 37 °C overnight. The DNA was purified by the PEG precipitation and CIA extraction (twice). The resultant 84-bp unit DNA was dephosphorylated by CIAP (Takara, 2250B) at 1 unit/nmol of DNA end at 37 °C overnight. Then, CIAP was added at 0.5 units/nmol of DNA end and the reaction mixture was further incubated at 37 °C overnight. The DNA was purified by the phenol/CIA extraction (twice) and ethanol precipitation. The DNA was digested by EcoRI (Takara, 1040A) at 30 units/nmol of EcoRI site at 37 °C overnight. The resultant 73-bp cohesive DNA was purified by ion-exchange chromatography using a TSKgel DEAE-5PW column (TOSOH BIOSCIENCE, 0007574; 250–550 mM NaCl gradient). After purification by ethanol precipitation, the 73-bp DNA was ligated to each other by using *E.coli*. DNA ligase (Takara, 2160A) at 1 unit/μg of DNA fragment at 16 °C overnight. The resultant palindromic 146-bp nucleosome DNA was purified by ethanol precipitation.

Histone octamers and the 180-bp or 146-bp palindromic DNA were mixed at a 1:1.1 molar ratio in 10 mM Tris-HCl buffer (pH 7.6) containing 2 M KCl, 1 mM EDTA, and 5 mM DTT. The solution was placed in a dialysis membrane bag (Spectrum, MWCO 6–8 kDa, cat. no. 132653) and dialyzed against the same buffer for 4 hrs at 4 °C. The concentration of KCl was then gradually decreased by diluting for 30 hrs with 10 mM Tris-HCl buffer (pH 7.6) containing 1 mM EDTA and 1 mM DTT using a peristaltic pump (ATTO, SJ-1211II-H). The sample was centrifuged for 10 min at 15,000 × *g* at 4 °C and the supernatant containing the reconstituted nucleosomes was incubated at 55 °C for 2 hrs and stored at 4 °C. Reconstituted nucleosomes were purified on a 6% native polyacrylamide gel (acrylamide:N,N′-methylenbisacrylamide = 59:1), using a Model 491 Prep Cell apparatus (Bio-Rad, 1702928). The eluted fractions were concentrated in Amicon Ultra-15 centrifugal filter units (Merck Millipore, 10 kDa MWCO). For cryo-EM analysis, the reconstituted nucleosomes were dialyzed against 20 mM HEPES-NaOH buffer (pH 7.2) containing 150 mM NaCl.

## Immunoblot analysis

The histones containing residue-specific acetylations or the nucleosomes used for the acetyltransferase activity assay were electrophoresed in a 10–20% SDS polyacrylamide gel (DRC, NXV-396HP20) and transferred onto a nitrocellulose membrane (BIO-RAD, 1620112) at 20 V for 10 min by the semi-dry method. The membrane was blocked with Bullet Blocking One for Western Blotting (Nacalai Tesque, 13779-01) for 10–30 min at 25 °C. Membranes were incubated for 20–40 min at 25 °C with Bullet ImmunoReaction Buffer (Nacalai Tesque, 18439-85) containing the following antibodies at the indicated dilution rate: H2AK5ac (Abcam, ab45152, 1/3000), H2B (Cell Signaling, 12364, 1/1000), H2BK12ac (Abcam, ab40883, 1/500), H2BK15ac (Abcam, ab62335, 1/500), H2BK16ac (Abcam, ab177427, 1/1000), H2BK20ac (Abcam, ab177430, 1/500), H2BK23ac (Abcam, ab222770, 1/1000), H3 (Merck, 07-690, 1/3000), H3K14ac (Merck, 07-353, 1/1000), H3K18ac (Abcam, ab1191, 1/1000), H3K23ac (Merck, 07-355, 1/1000), H3K27ac (Merck, 07-360, 1/3000), H4 (Abcam, ab10158, 1/1000), H4K5ac (MABI, 0405, 1/500), H4K8ac (MABI, 0408, 1/500), H4K12ac (MABI, 0412, 1/500), or H4K16ac (MABI, 0416, 1/500). The membranes were then washed with TBS-T (four times, for 5 min each time) and incubated with Bullet ImmunoReaction Buffer (Nacalai Tesque, 18439-85) containing HRP-conjugated anti-rabbit IgG (Cytiva, NA934, 1/10000) or anti-mouse IgG (Cytiva, NA931, 1/10000) for 20 min at 25 °C. The membranes were then washed with TBS-T (four times, for 5 min each time) and were detected using enhanced chemiluminescence (Chemi-Lumi One Super: Nacalai Tesque, 02230-30). The immunoblotted membranes were imaged using ImageQuant LAS-4000 (Cytiva). From the images obtained, the immunoblotting signal intensity of each band was quantified using ImageJ 1.53 (RSB, https://imagej.nih.gov), and the background intensity was subtracted. Data were analyzed using Microsoft 365 (Excel). Uncropped gels of SDS-PAGE and immunoblotting are shown in Supplementary Fig. 22.

## Acetyltransferase assay

Acetyltransferase activity assays of p300$_{BRPHZT}$ against the nucleosome were performed at a molar ratio of p300$_{BRPHZT}$:180-bp nucleosome:acetyl-CoA (Nacalai Tesque, 00546-54) = 1:1:10 in 10 μl of reaction solution containing 5.6 mM HEPES-NaOH (pH 7.4), 4.4 mM Tris-HCl (pH 7.4), and 130 mM NaCl. Briefly, in a 0.2-ml PCR tube (Thermo Fisher Scientific, 3414JP), 2.4 μl of 10 mM HEPES-NaOH (pH 7.4) buffer containing 10 pmol p300$_{BRPHZT}$ and 300 mM NaCl was mixed with 3.2 μl of 10 mM HEPES-NaOH (pH 7.4) buffer containing 10 pmol nucleosomes (with or without the indicated residue-specific histone acetylations) and 150 mM NaCl, and 4.4 μl of 10 mM Tris-HCl (pH 7.4) buffer containing 100 pmol acetyl-CoA. Every minute for 0–3 min after the reaction at 37 °C, each sample was mixed with 20 μl of 4×SDS-PAGE loading buffer and immediately heated at 95 °C for 3 min to stop the reaction, and 30 μl of water was added. For each sample solution, 4 μl was electrophoresed in a 10–20% SDS Tris-Tricine polyacrylamide gel and applied for the immunoblot analysis. All assays were independently repeated three times.

## Cryo-EM sample and grid preparation

To prepare p300$_{BRPHZT}$·H4acNuc and CBP$_{BRPHZT}$·H4acNuc complexes, p300$_{BRPHZT}$ (or CBP$_{BRPHZT}$), acetyl-CoA, and H4acNuc were mixed at a molar ratio of 4:6:1 in 20 mM HEPES-NaOH buffer (pH 7.2) containing 150 mM NaCl, 1 μM ZnCl$_2$, and 1 mM TCEP and incubated for 30 min. To purify and crosslink the complex, the reaction mixture was fractionated by the GraFix method[81]. A gradient for GraFix was formed with a top solution of 10 mM HEPES-NaOH buffer (pH 7.2) containing 50 mM NaCl, 1 mM TCEP, 10% glycerol, and 0.01% glutaraldehyde) and a bottom solution of 10 mM HEPES-NaOH buffer (pH 7.2) containing 50 mM NaCl, 1 mM TCEP, 30% glycerol, and 0.15% glutaraldehyde, using a Gradient Master (BioComp). The reaction mixture was applied onto the top of the gradient solution and was centrifuged at 288,000 × *g* at 4 °C for 17 hrs using an SW41 Ti rotor (Beckman Coulter). The gradient was fractionated using a Piston Gradient Fractionator (BioComp). A portion of each fraction was analyzed by electrophoretic mobility shift assay. Electrophoresis was performed at 150 V for 50 min on 6% native TBE polyacrylamide gels, and then the fractions corresponding to H4acNuc complex were pooled. Glutaraldehyde was quenched by the addition of Tris-HCl at pH 7.6 to a final concentration of 100 mM. The samples were dialyzed against 10 mM HEPES-NaOH buffer (pH 7.2) containing 50 mM NaCl and 1 mM TCEP and concentrated using Amicon Ultra centrifugal filter units (Merck Millipore, 100 kDa MWCO) for cryo-EM analyses. A 3.0-μl aliquot of the samples of the p300$_{BRPHZT}$·H4acNuc complex (2.0 mg/ml) or the CBP$_{BRPHZT}$·H4acNuc complex (2.0 mg/ml) was each applied to glow-discharged, holey, copper grids (Quantifoil Cu R1.2/1.3, 300 mesh) with a thin carbon-supported film. The grids were plunge-frozen into liquid ethane using Vitrobot Mark IV (Thermo Fisher Scientific). Parameters for plunge-

freezing were set as follows: blotting time, 3 sec; waiting time, 3 sec; blotting force, −5; humidity, 100%; and chamber temperature, 4 °C.

## Cryo-EM data collection and image processing

Cryo-EM data were collected with a Tecnai Arctica transmission electron microscope (Thermo Fisher Scientific) operated at 200 kV using a K2 summit direct electron detector (Gatan) at a nominal magnification of 23,500× in electron-counting mode, corresponding to a pixel size of 1.47 Å per pixel. The movie stacks were acquired with a defocus range of −0.9 to −1.7 μm with total exposure time of 12 s fragmented into 40 frames with a dose rate of 50.0 e⁻/Å². Automated data acquisition was carried out using SerialEM software (v3.8)[82]. The cryo-EM data were also collected with a Krios G4 transmission electron microscope (Thermo Fisher Scientific) operated at 300 kV using a K3 direct electron detector (Gatan) at a nominal magnification of ×105,000 in electron-counting mode, corresponding to a pixel size of 0.83 Å per pixel. The movie stacks were acquired with a defocus range of −0.8 to −2.0 μm with a total exposure time of 2.3 sec fragmented into 50 frames with the dose rate of 50.0 e⁻/Å². These data were automatically acquired by the image-shift method using the EPU software (v2.9). All cryo-EM experiments were performed at the RIKEN Yokohama cryo-EM facility. All image processing was performed with RELION-3.1[83]. Dose-fractionated image stacks were subjected to beam-induced motion correction using MotionCor2[84] and the CTF parameters were estimated with CTFFIND-4.1[85]. Particles were automatically picked using crYOLO (v1.7.6)[86] with a box size of 135 × 135 pixels for the Tecnai Arctica dataset and a box size of 225 × 225 for the Krios G4 dataset. These particles were extracted and subjected to several rounds of 2D and 3D classifications using RELION-3.1. The selected particles were then re-extracted and subjected to 3D refinement, Bayesian polishing[87], and subsequent postprocessing of the map improved its global resolution, according to the Fourier shell correlation with the 0.143 criterion[88]. Details of the data collection and image processing are summarized in Supplementary Table 3 and Supplementary Figs. 4, 5, 6, 8.

## Model building and refinement

For p300$_{BRPH}$, each domain of the p300$_{BRPH}$ crystal structure (PDB ID: 6GYR) was divided and fitted to a cryo-EM map as a rigid body using fit in map in the visualization software UCSF Chimera (v1.15)[89]. For CBP$_{BRPH}$, as with p300, each domain of the crystal structure (BD and HAT; PDB ID: 5U7G)[90] and AlphaFold structure (v2.2.2; AF-Q92793-F1) was fitted to the cryo-EM map. For H4acNuc, the crystal structure of the 146-bp nucleosome (PDB ID: 1KX3)[91] was fitted to a cryo-EM map, and then linker DNA and H4NTac were manually modeled and each histone was substituted for an amino acid residue using Coot-0.9.8.1[92]. For all structures, when rigid bodies could not be fitted to the cryo-EM map, we performed a flexible fitting by using the plug-in ISOLDE 1.4[93]. The final model was refined by PHENIX (v1.18.2)[94], and the stereochemistry was assessed by MolProbity[95]. Statistics for cryo-EM model refinement are summarized in Supplementary Table 3. All figures were generated using either UCSF Chimera (v1.15)[89], UCSF ChimeraX (v1.4)[96], or PyMOL (v2.5)[97]. The mapping of electrostatic potential was achieved using PyMOL with the Adaptive Poisson-Boltzmann Solver (APBS) Electrostatics plug-in (https://pymolwiki.org/index.php/APBS_Electrostatics_Plugin).

## Microscale thermophoresis

The wild-type p300$_{BRP}$ and 4A-substituted mutant were each fluorescently labeled, using a His tag Labeling Kit (NanoTemper Technologies, MO-L018). For measurements using CBP30, CBP30 was added beforehand to the wild-type and the 4A-substituted mutant to a final concentration of 10 μM and incubated for at least 20 min. Labeled proteins were mixed with 2-fold serial dilutions of the 180-bp unmodified or H4NT-acetylated nucleosome in 10 mM HEPES-NaOH buffer

(pH 7.4) containing 150 mM NaCl and 0.05% Tween-20. Measurements were taken by using a Monolith NT.115 Instrument (NanoTemper Technologies) at 25 °C according to the manufacturer's instructions. Each assay was independently repeated three times using the unmodified and H4NT-acetylated nucleosome. The measured data were fitted to the Hill equation using MO. Affinity analysis software v2.3 (NanoTemper Technologies).

## Isothermal titration calorimetry

H2B (residues 1–27, 8–20, and 16–27) and H4 (1–20) peptides with indicated Kac(s) were purchased from Toray Research Center. Measurements were conducted at 25 °C in 10 mM HEPES-NaOH buffer (pH 7.4) containing 150 mM NaCl on a MicroCal Auto-iTC$_{200}$ microcalorimeter (Malvern). Approximately 400 μl of 100 μM protein solution was loaded into the sample cell, and each of 1 mM acetylated histone NT peptides was loaded into an injection syringe. The titration consisted of 20 injections; the first injection of 0.4 μl and the remaining 19 injections of 2.0 μl were performed with an interval of 150 sec, a reference power of 5 μCal/sec, and 750 rpm stir speed during the titration. Each experiment was independently repeated twice. The collected data were analyzed using Origin 7 SR4 v7.0552 software (OriginLab Corporation) supplied with the instrument to calculate enthalpies of binding (ΔH) and $K_D$. A single binding site model was used in all assays.

## Nucleosome thermostability assay

The thermal stability of the 180-bp nucleosome with specific residues acetylated and of the unmodified nucleosome was measured using a QuantStudio 6 PCR system (Life Technologies). In 20 μl of 20 mM Tris-HCl buffer (pH 7.6) containing 1 mM EDTA and 1 mM dithiothreitol, 20 pmol nucleosomes were reacted with a 4-fold concentration of Protein Thermal Shift Dye (Life Technologies). Fluorescence intensity produced by the reaction was monitored ($\lambda_{ex}$ = 580 nm; $\lambda_{em}$ = 623 nm) every second from 25.0 °C to 99.9 °C with a temperature change of 0.015 °C/sec. Data were analyzed using Applied Biosystems Protein Thermal Shift Software v1.2. Three independent data sets were obtained and averaged. Data were normalized with the maximal fluorescence intensity as 100%.

## Mass spectrometry

180-bp nucleosomes reconstituted in the presence (H4K12ac+K16ac) or absence (unmod) of H4K12ac/K16ac were reacted with wild-type p300$_{BRPHZT}$ and acetyl-CoA in a 1:1:10 molar ratio in 10 μl of solution. The solution before or 1 or 3 min after the reaction was heat-denatured at 95 °C for 3 min and sampled. Each sample was separated by SDS-PAGE and stained with CBB. Separated protein bands were cut out and de-stained, followed by in-gel digestion with *Achromobacter* protease I (a gift from Dr. T. Masaki, Ibaraki University, Ibaraki, Japan). The resulting peptides were analyzed using LC-MS/MS consisting of Q Exactive (Thermo Fisher Scientific) and Easy nLC1000 (Thermo Fisher Scientific). For liquid chromatography, 0.1% formic acid was used as solvent A and 80% acetonitrile containing 0.1% formic acid was used as solvent B. The peptides were separated on a spray column (Nikkyo Technos, NTCC-360/75-3-105) with a 0–44% B solvent gradient for 20 min. Measurements were technically repeated three times and carried out in positive mode using the TOP10 method. The acquired data were analyzed with MASCOT 2.8 and Proteome Discoverer 3.0. The MASCOT search was performed with the following settings: database; Swiss-Prot (Homo sapiens) and an in-house database including human histones H2A, H2B, H3.1, and H4; parameters: enzyme = trypsin/P; maximum missed cleavages = 4; variable modifications = Gln- > pyro-Glu (N-terminal Q), Oxidation (M), Acetyl (K); product mass tolerance = ±30 ppm; product mass tolerance = ±30 mmu; instrument type = ESI-TRAP. Comparison of acetylated peptide abundance in unmod and K12ac+K16ac over time was carried out by

calculating [(acetylated peptide abundance in H4K12ac+K16ac)/(protein abundance in H4K12ac+K16ac)]/[(acetylated peptide abundance in unmod)/(protein abundance in unmod)]. Acetylated peptide abundance was plotted based on peak area using Qual Browser (Thermo Fisher Scientific). Histone protein abundance was determined from Proteome Discoverer 3.0. Positions of Kac in p300$_{BRPHZT}$ were determined by LC-MS/MS. The protein band was digested in gel with trypsin. Peptides were identified in the same manner as histones were analyzed.

## Multiple sequence alignment

The multiple sequence alignment was performed by Clustal W 2.1[98] in UniProt (http://www.uniprot.org/align/) and was rendered by ESPript 3.0[99] (http://espript.ibcp.fr/ESPript/ESPript/). Amino acid sequences used for the alignment are: human p300 (UniProt ID: Q09472), human CBP (Q92793), human PCAF (Q92831), human GCN5 (Q92830), human BRD4 (O60885), human TAF1 (P21675), *D. melanogaster* Nejire/dCBP (Q9W321), *C. elegans* CBP-1 (P34545), and *A. thaliana* PCAT2 (Q9C5X9). Amino acid sequences of other human bromodomain-containing proteins in Supplementary Fig. 13 were obtained from UniProt.

## Reporting summary

Further information on research design is available in the Nature Portfolio Reporting Summary linked to this article.

## Data availability

Data supporting this study, including Supplementary Information and Source Data, are available in the article. The cryo-EM density maps have been deposited in the Electron Microscopy Data Bank (EMDB, www.ebi.ac.uk/pdbe/emdb/) under the accession codes EMD-34588, EMD-34589, EMD-34590, EMD-34591, EMD-34592, EMD-34593, EMD-34594, EMD-34595, EMD-34596, and EMD-34597. The atomic coordinates have been deposited in the Protein Data Bank (PDB, www.rcsb.org) under the accession codes 8HAG, 8HAH, 8HAI, 8HAJ, 8HAK, 8HAL, 8HAM, and 8HAN. Structural models used in this study can be found in PDB under the accession codes 1KX3, 5LKU, 5U7G, 6GYR, and 7W9V, and AlphaFold Protein Structure Database under the accession code AF-Q92793-F1. The mass spectrometry proteomics data have been deposited with the ProteomeXchange Consortium via the PRIDE partner repository[100] with the dataset identifier PXD040835. Source data are provided with this paper.

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

## Acknowledgements

We thank the RIKEN Yokohama cryo-EM facility of the Center for Biosystems Dynamics Research (BDR) for their support in the cryo-EM data collection; Kazuharu Hanada, Mio Inoue, Sayako Miyamoto-Kohno, and Mie Goto for sample preparation; Nando Dulal Das, Hideaki Niwa, Shinsuke Ito, Haruhiko Koseki, and Chunaram Choudhary for discussions; Yuki Saito for clerical assistance; and Masami Horikoshi, Shigeyuki Yokoyama, and Minoru Yoshida for encouragement. This work was supported by Grants-in-Aid from the Japan Society for the Promotion of Science under Grant Numbers JP19K16062, JP21K15035 [to M.K.], JP16H05089, JP20H03388, JP20K21406, and JP21H05764 [to T.U.]; the PRESTO program of the Japan Science and Technology Agency under Grant Number JPMJPR12A3 [to T.U.]; the Platform Project for Supporting Drug Discovery and Life Science Research (BINDS) from Japan Agency for Medical Research and Development under Grant Numbers JP21am0101082 [to M.S.] and JP21am0101115: support No. 2959; the Structural Cell Biology Project of RIKEN BDR [to M.K.]; the Epigenome Manipulation Project of the All-RIKEN Projects [to T.U.]; and the RIKEN Pioneering Project, Genome Building from TADs [to T.U.].

## Author contributions

M.K. designed the research, performed the cryo-EM structure and biochemical analyses, and drafted the structural part of the manuscript. S.M., M.W., and S.S. performed the biochemical analysis. T.U-K. assisted the cryo-EM measurement. T.S. and N.D. performed the mass spectrometric analysis. M.S. assisted the protein preparation. T.U. conceived and supervised the project, designed the research, and wrote the manuscript.

## Competing interests

The authors declare no competing interests.
