## [Peer Review File · Nature Communications]

Epigenetic mechanisms to propagate histone acetylation by p300/CBPREVIEWER COMMENTS

Reviewer #1 (Remarks to the Author):

This manuscript presents cryo-electron microscopy structures of p300/CBP bound to the nucleosome with histone H4 N-terminal tail (NT) pre-acetylated (H4K12ac/K16ac). The structures provide insights into how p300/CBP acetylates non-H4 histone NTs within the same nucleosome. They propose p300/CBP 'replicates' histone NT acetylation within the H3-H4 tetramer to inherit epigenetic storage, and 'transcribes' it from the H3-H4 tetramer to the H2B-H2A dimers to activate context-dependent gene transcription through local nucleosome destabilization.

The study presented in the manuscript provides high-quality structural and biochemical data. It nicely illustrates the process of how the acetylated H4 N-terminal tail facilitates the acetylation of other histones by p300/CBP. Overall, the manuscript is well written. The content should be interesting to people in the field of epigenetics and chromatin. I have two related comments:

1. As indicated by the authors, structures of the nucleosome bound to p300/CBP have been determined previously and used to explain how p300/CBP work (reference 23). The authors need to compare their structures with earlier ones to show why their new structures are more important.
2. Is it known that in vivo H4 K12 and K16 are acetylated before the acetylation of the other sites? Is it necessary for p300/CBP to bind the pre-acetylated H4 k12 and K16 to acetylate other sites/histones? The authors listed reference 13 as the basis for their study. This needs to be elaborated for readers who are not experts on this topic. It is not obvious to reach such a conclusion by reading reference 13.

Reviewer #2 (Remarks to the Author):

Kikuchi et al. have structurally and biochemically analysed how p300 acetylates a single nucleosome. This is important work, as nucleosome acetylation regulates enhancer mediated transcriptional regulation, which is crucial for diverse cellular responses. The authors used the physiological pre-acetylated nucleosome substrate and verified its activity. Interestingly, they determined multiple structures of the p300 H4acNuc complex, which not only shows how p300 recognizes H4acNuc, but also shows how the HAT domain becomes dynamically repositioned to support acetylation of various histone tails. The authors identify a p300-DNA interaction and show that mutation of the DNA binding motif impacts binding of H4acNuc and its acetylation activity. Moreover, the authors use nucleosome with various pre-acetylation to uncover a distance code that explain how multi-site acetylation is regulated. Finally, the authors looked at a H2 destabilisation effect of H2B acetylation, which indicates a thermostability. If this destabilisation has functional relevance is not yet clear. In general, the study is very well executed and I strongly support its publication, once a few points have been addressed.

Major points:

The authors look at the distance of the p300 modified tails to the active site and suggests specific modification patterns. Can these be verified by mass-spec to ask whether p300 has indeed the predicted preference?

It is interesting that the authors identified that multisite H2BNTac affects nucleosome stability. However, I wonder what physiological impact this has on histone exchange at room temperature. Can the authors use other histone destabilisation assays to confirm the result and justify that the 2-degree change in thermal stability is significant? It might be also useful to include a model that shows the relevance of the data. Otherwise I would suggest to tone down the conclusions.

I like that the authors suggest that H2BNTac is most specific for p300 in vitro – could this be verified in vivo?

In order to judge the quality of the maps I tried to access the data on EMDB, but the EM maps have not been deposited and are not visible.

The data suggest that the N-terminal tails are visible in the structure, but the data quality is unclear. Therefore, the authors should provide zoomed in local resolution and atomic model views of the histone tails from different maps, as this would help the reader understand the resolution from which major conclusions in the study are made.

Supplementary Fig4.f) Please display all FSC curves for all models. Only 2 are shown.

Minor point:

Describe the p300 section (e.g. aa1-1047 & 1837-2414) that were left out and justify why these will not alter the results.

The data in Figure 1b and supplementary Figure 2 showing H2BK16ac appear not to fit – in Fig. 1B less activity is shown for the unmodified H4 than in the supplementary figure and the activity levels overall are different.

It is difficult to see this in the supplementary figures: “Besides p300H2B·H4acNuc, there were several similar complexes at low resolution in which H2BNT was located near the substrate-binding pocket of HAT (Supplementary Fig. 4, 5).” Could the authors better indicate this in the figure. Also, the authors should mention that they detected K12,K16 and K20 acetylation (supplementary Figure 2).

Fig2A – please mark the histone that is in proximity to the HAT and label it – this is difficult to see in the figure.

In the discussion (page 10, line 297) the epigenetic storage and mitotic bookmarks should be introduced more.

I like the model presented on page 10, line 298ff. Could the authors draw a model to illustrate this?

Response to the reviewers' comments

To Reviewer 1

This manuscript presents cryo-electron microscopy structures of p300/CBP bound to the nucleosome with histone H4 N-terminal tail (NT) pre-acetylated (H4K12ac/K16ac). The structures provide insights into how p300/CBP acetylates non-H4 histone NTs within the same nucleosome. They propose p300/CBP 'replicates' histone NT acetylation within the H3-H4 tetramer to inherit epigenetic storage, and 'transcribes' it from the H3-H4 tetramer to the H2B-H2A dimers to activate context-dependent gene transcription through local nucleosome destabilization.

The study presented in the manuscript provides high-quality structural and biochemical data. It nicely illustrates the process of how the acetylated H4 N-terminal tail facilitates the acetylation of other histones by p300/CBP. Overall, the manuscript is well written. The content should be interesting to people in the field of epigenetics and chromatin. I have two related comments:

1. As indicated by the authors, structures of the nucleosome bound to p300/CBP have been determined previously and used to explain how p300/CBP work (reference 23). The authors need to compare their structures with earlier ones to show why their new structures are more important.

[Response]

We thank this reviewer for their favorable evaluation. As suggested, we have included a new figure that compares the previously published structure (PDB ID: 7W9V; new reference 28) with ours and explained the differences as follows (new Supplementary Figure 11; lines 196–207).

[Revised text in Results]

Cryo-EM maps of the histone tails with HAT of p300_{BRPH} or CBP_{BRPH} in close proximity in each structure are superimposed on the corresponding structural model (Supplementary Fig. 10). For each of these complexes, a structural comparison with the recently reported p300·nucleosome complex structure¹ is shown in Supplementary Figure 11. In the most resolved structure by Hatazawa *et al.*¹ of catalytically inactive p300(BRPH_{ΔAILZ}) complexed with the nucleosome core particle reconstituted with a 145-bp Widom 601 sequence (complex I; PDB ID: 7W9V), its HAT and bromodomain contact nucleosomal DNA at SHLs 2 and 3. p300 in their structure does not 'read/write' Kac on the nucleosome because bromodomain is not in close proximity to any histone NTs, although the catalytic site of HAT is near one of the H4NTs. Indeed, the position of p300 on the nucleosome in their structure is quite different from any of our structures in which p300 presumably 'reads/writes' Kac on the nucleosome.

(References are attached at the end of this document.)

2. Is it known that in vivo H4 K12 and K16 are acetylated before the acetylation of the other sites? Is it necessary for p300/CBP to bind the pre-acetylated H4 k12 and K16 to acetylate other sites/histones? The authors listed reference 13 as the basis for their study. This needs to be elaborated for readers who are not experts on this topic. It is not obvious to reach such a

conclusion by reading reference 13.

[Response]

We appreciate this important point of view. It is difficult to state precisely whether K12 and K16 of H4 are acetylated before the acetylation of other lysine residues *in vivo* because it depends on the literature and to which residues we compare them. However, it is known that acetylation of five lysine residues in the N-terminal tail of H4 or H3, including H4K12ac and H4K16ac, to which p300 preferentially binds (*i.e.*, H4K8ac, H4K12ac, H4K16ac, H3K14ac, and H3K18ac) is all stably maintained in mitotic chromatin (Behera *et al. Cell. Rep.* 27, 400, 2019). This evidence suggests that these five residues are already acetylated from early G₁ phase just after mitosis. In the Introduction, we have added this information briefly and smoothly connected it to the logic of our structural studies that focus on H4 K12ac and K16ac (lines 62–64). Also, we have elaborated on this point in the Discussion and introduced previous findings on the maintenance of K8ac, K12ac, and K16ac in histone H4. Accordingly, we have revised our discussion of a possible p300/CBP-mediated self-perpetuation mechanism for H4NTac (lines 340–359).

[Revised text in Introduction]

Interestingly, the Kac 'reader' bromodomain of p300 is known to preferentially bind to H4NTac containing both K12ac and K16ac (H4K12ac/K16ac) that are stably maintained on mitotic chromatin^{2, 3}. In this study, we report cryo-EM structures revealing how the p300/CBP multidomain involved in the 'read/write' of histone acetylation recognizes H4NTac containing H4K12ac/K16ac and acetylates non-H4 histone NTs in the same nucleosome.

[Revised text in Discussion]

To ensure the self-perpetuation of Kac in the H3-H4 tetramer, it would be important for p300/CBP to preferentially bind to nucleosomes containing H4K12ac/K16ac ($K_D = 25 \mu\text{M}$), H4K8ac/K12ac ($K_D = 90 \mu\text{M}$), or H3K14ac/K18ac ($K_D = 104 \mu\text{M}$) *via* bromodomain². Indeed, acetylation of these five residues (*i.e.*, H4K8ac, H4K12ac, H4K16ac, H3K14ac, and H3K18ac) is stably maintained during mitosis in mammalian cells³, suggesting that they are already acetylated in the early G₁ phase just after mitosis. For H4NTac in particular, H4K8ac is a mitotic bookmark for GAGA pioneer factor to activate its target genes in fly⁴. H4K12ac, which is recognized by the bromodomain pocket of p300_{BRPH}, is already acetylated in the cytoplasm^{5, 6, 7} and maintained during mitosis in living cells⁸. Also, H4K16ac is intergenerationally maintained in fly and mammals, making it instructive for future gene activation⁹. However, our data and those of another group¹⁰ show that p300 can hardly acetylate nucleosomal H4 at K16 and not much at K12. Given the distance between the H4NT pair, these data suggest that H4K8ac/K12ac self-perpetuates by p300/CBP with the help of H3K14ac/K18ac and cytoplasmic acetylation of H4K12. Self-perpetuation of H4K16ac presumably requires the MYST (MOZ, YBF2, SAS2, and Tip60)-type histone acetyltransferases, such as MOF/KAT8^{11, 12}. For H3NTac, the distance between the H3NT pair suggests that H3K14ac/K18ac may self-perpetuate by p300/CBP alone (Supplementary Fig. 18b). Such p300/CBP-mediated self-perpetuation of Kac in the H3-H4 tetramer would contribute to both the inheritance of Kac information to daughter cells and the p300/CBP-driven rapid transcriptional activation of specific genes immediately after mitosis¹³.

To Reviewer 2

Kikuchi et al. have structurally and biochemically analysed how p300 acetylates a single nucleosome. This is important work, as nucleosome acetylation regulates enhancer mediated transcriptional regulation, which is crucial for diverse cellular responses. The authors used the physiological pre-acetylated nucleosome substrate and verified its activity. Interestingly, they determined multiple structures of the p300 H4acNuc complex, which not only shows how p300 recognizes H4acNuc, but also shows how the HAT domain becomes dynamically repositioned to support acetylation of various histone tails. The authors identify a p300-DNA interaction and show that mutation of the DNA binding motif impacts binding of H4acNuc and its acetylation activity. Moreover, the authors use nucleosome with various pre-acetylation to uncover a distance code that explain how multi-site acetylation is regulated. Finally, the authors looked at a H2 destabilisation effect of H2B acetylation, which indicates a thermostability. If this destabilisation has functional relevance is not yet clear. In general, the study is very well executed and I strongly support its publication, once a few points have been addressed.

Major points:

The authors look at the distance of the p300 modified tails to the active site and suggests specific modification patterns. Can these be verified by mass-spec to ask whether p300 has indeed the predicted preference?

[Response]

We appreciate the reviewer's numerous comments that have helped us to improve this study. As suggested, we have performed a mass spectrometric analysis to verify the patterns (new Supplementary Table 2). The data were in good agreement with our original conclusions drawn from structural and immunoblot analyses of acetylation target residues, suggesting that pre-acetylation of the H4 N-terminal tail at K12 and K16 indeed promotes p300_{BRPHZT}-catalyzed acetylation of the H2B N-terminal tail most among the four histone N-terminal tails. We have added these findings to the revised manuscript (lines 102–109).

[Revised text in Results]

Next, we performed mass spectrometry to comprehensively detect histone residues whose acetylation by p300_{BRPHZT} is facilitated in the presence of H4K12ac/K16ac (Supplementary Table 2). The results showed that at the one-minute post-reaction timepoint, the presence of H4K12ac/K16ac increased the amount of protease-digested peptides containing multisite H2BNTac of K11ac, K12ac, K15ac, K16ac, or K20ac by 4.3- to 46-fold. The amount of protease-digested peptides containing multisite H3NTac of K14ac, K18ac, K23ac, or K27ac also increased 1.5- to 11-fold. These p300_{BRPHZT}-catalyzed acetylation preferences were in good agreement with the results of immunoblot analysis.

(References are attached at the end of this document.)

It is interesting that the authors identified that multisite H2BNTac affects nucleosome stability. However, I wonder what physiological impact this has on histone exchange at room temperature. Can the authors use other histone destabilisation assays to confirm the result and justify that the 2-degree change in thermal stability is significant? It might be also useful to include a model that shows the relevance of the data. Otherwise I would suggest to tone down

the conclusions.

[Response]

Unfortunately, due to the lack of histone chaperones and chromatin remodeling factors, we were unable to examine the impact of the multisite H2BNTac on histone exchange. Therefore, we have stated in the Discussion that this point is one of the limitations of our study and needs to be verified in the future (lines 393–397).

Nevertheless, a study by Ito et al. (*Genes Dev* 14, 1899, 2000) showed that transcriptional activator (Gal4–VP16)–dependent histone acetylation by p300 facilitates the transfer of the H2A-H2B dimers from the nucleosomes to the histone chaperone NAP-1, possibly through its interaction with acetylated H2A-H2B. This evidence, cited as reference 37 in the original manuscript, supports our results that multisite H2BNTac promotes the dissociation of the H2A-H2B dimer from the nucleosome in terms of thermal stability and our discussion that this phenomenon may promote the dimer exchange by histone chaperones under physiological conditions. We have therefore discussed this point in the Discussion (lines 382–384; reference 66).

[Revised text in Discussion]

Importantly, TF-dependent histone acetylation by p300 facilitates the transfer of the H2A-H2B dimers from the nucleosomes to the histone chaperone NAP-1, possibly *via* its interaction with acetylated H2A-H2B¹⁴. In this light and given other findings on the behavior of H2B^{15, 16, 17}, it makes sense that p300-catalyzed H2BNTac is the genuine signature of active enhancers and their target promoters^{18, 19}. In addition, RNAPII complexed with the histone chaperone FACT flips the histone octamer and exchanges one H2A-H2B dimer during traverse across the nucleosome²⁰. Thus, H2BNTac information ‘written’ by p300/CBP should be ‘erased’ by subsequent successive transcription, and the H2BNTac information that guides transcription should be rapidly and repeatedly updated. H3NTac-activated histone exchange from H2A to H2A.Z on chromatin²¹ may further facilitate nucleosomal destabilization and RNAPII recruitment at promoters through p300-driven multisite acetylation of H2A.Z²². It should be noted, however, that our study does not make clear how the thermal instability of the nucleosome caused by the p300-catalyzed multisite H2BNTac, *per se*, affects histone exchange under physiological conditions, nor whether p300 acetylates H2BNT most in the cell by the present structural mechanism. These points are limitations of this study and need further biochemical and cell biological validation.

I like that the authors suggest that H2BNTac is most specific for p300 *in vitro* – could this be verified *in vivo*?

[Response]

It was difficult with our current technology to validate the above biochemical data *in vivo*. In the revised manuscript, we have stated this point as another limitation of our study (lines 393–397).

[Revised text]

The same as above.

In order to judge the quality of the maps I tried to access the data on EMDB, but the EM maps have not been deposited and are not visible.

[Response]

We have attached validation reports for all cryo-EM structures which have been deposited in the Electron Microscopy Data Bank and the Protein Data Bank. To show the quality of the maps, we have demonstrated the cryo-EM densities of the histone tails with the HAT domain of p300_{BRPH} or CBP_{BRPH} in close proximity in each structure and the superposition on the corresponding structural models (new Supplementary Figure 10; lines 196–198).

[Revised text in Results]

Cryo-EM maps of the histone tails with HAT of p300_{BRPH} or CBP_{BRPH} in close proximity in each structure are superimposed on the corresponding structural model (Supplementary Fig. 10).

The data suggest that the N-terminal tails are visible in the structure, but the data quality is unclear. Therefore, the authors should provide zoomed in local resolution and atomic model views of the histone tails from different maps, as this would help the reader understand the resolution from which major conclusions in the study are made.

[Response]

As with the above response, we have added a new Supplementary Figure 10, which shows the local cryo-EM densities of the zoomed N-terminal tails each superimposed on the corresponding atomic model.

[Revised text in Results]

The same as above.

Supplementary Fig4.f) Please display all FSC curves for all models. Only 2 are shown.

[Response]

As suggested, we have shown Fourier shell correlation curves for all models (Supplementary Figures 4f, 5c, and 7f).

Minor point:

Describe the p300 section (e.g. aa1-1047 & 1837-2414) that were left out and justify why these will not alter the results.

[Response]

We have described the fact that our p300 construct lacks the N- and C-terminal regions. This construct was designed based on previous reports suggesting that similar N/C-truncated forms of human p300 can ‘read/write’ histone acetylation in the nucleosomes. Therefore, we have added this information as a reason for selecting residues 1048–1836. We also confirmed by Western blot (new Supplementary Fig. 1c) and mass spectrometry (new Supplementary Table 1) that our p300 construct (residues 1048–1836) underwent autoacetylation, similar to already known p300 protein constructs with catalytic activity. We have added this data to the Results (lines 74–85).

[Revised text in Results]

To understand how p300/CBP ‘reads/writes’ histone acetylation, we expressed human p300_{BRPHZT} protein (residues 1048–1836), containing bromodomain, RING, and PHD (BRP), the catalytically active HAT domain with the autoinhibitory loop (AIL), ZZ, and the TAZ2 domain (BRPHZT; Fig. 1a, Supplementary Fig. 1a, b), using a baculovirus expression system. The p300_{BRPHZT} construct, which lacks the N- and C-terminal regions, was designed based on reports suggesting that similar human p300 constructs (residues 1035–1830 and residues 965–1810) ‘read/write’ histone acetylation in the nucleosomes^{22, 23}. Similar to known catalytically active p300 protein constructs^{24, 25, 26, 27}, the purified p300_{BRPHZT} protein was confirmed to be an autoacetylated form containing K1542ac, K1546ac, K1549ac, K1550ac, K1551ac, K1554ac, K1555ac, K1558ac, and K1560ac that depends on its own catalytic activity (Supplementary Fig. 1c, Supplementary Table 1).

The data in Figure 1b and supplementary Figure 2 showing H2BK16ac appear not to fit – in Fig. 1B less activity is shown for the unmodified H4 than in the supplementary figure and the activity levels overall are different.

[Response]

The difference in results between the two data sets arose because the measurements were independent (*i.e.*, one minute post-reaction data with and without the p300 inhibitor CBP30 and 0–3 minutes time course data without that p300 inhibitor). For clarity, we have revised the corresponding text (lines 98–101).

[Revised text in Results]

Because the measurements shown in Figure 1b and Supplementary Figure 2 are independent, there are some differences, such as the degree of the H4K12ac/K16ac-dependent increase in H2BK16ac. However, both results are consistent in terms of statistical significance.

It is difficult to see this in the supplementary figures: “Besides p300H2B·H4acNuc, there were several similar complexes at low resolution in which H2BNT was located near the substrate-binding pocket of HAT (Supplementary Fig. 4, 5).” Could the authors better indicate this in the figure. Also, the authors should mention that they detected K12, K16 and K20 acetylation (supplementary Figure 2).

[Response]

We have included a figure showing close-up views of these low-resolution subcomplexes as new Supplementary Figure 4g. Along with newly performed mass spectrometry, we have described that we detected that p300_{BRPHZT} promotes acetylation of multiple lysine residues from K11 to K20 of H2B when H4NT is pre-acetylated (lines 145–148).

[Revised text in Results]

Besides p300_{H2B}·H4acNuc, there were several similar complexes at low resolution in which H2BNT was located near the substrate-binding pocket of HAT (Supplementary Fig. 4g). This is consistent with our biochemical results that p300_{BRPHZT} facilitated acetylation of multiple lysine residues from K11 to K20 of H2B when H4NT is pre-acetylated (Supplementary Table 2). Thus, it is likely that HAT of p300_{BRPH} can acetylate various lysine residues around K16 of H2BNT by a similar structural mechanism.

Fig2A – please mark the histone that is in proximity to the HAT and label it – this is difficult to see in the figure.

[Response]

We have colored and labeled the histones in proximity to each HAT domain in Figure 2a.

In the discussion (page 10, line 297) the epigenetic storage and mitotic bookmarks should be introduced more.

[Response]

We have completely revised the description in this paragraph regarding the propagation of histone acetylation to conform to your suggestion (lines 340–359). First, to avoid confusion, we have aligned the notation in this paragraph to ‘mitotic bookmark’ instead of using the term ‘epigenetic storage’. Next, based also on reviewer 1’s comments, we have discussed five acetylated lysine residues that are considered mitotic bookmarks and their maintenance mechanisms, for each of H4K8ac/K12ac, H4K16ac, and H3K14ac/K18ac. The term ‘epigenetic storage’ was used in the final paragraph of Discussion only to discuss H3/H4 acetylation in contrast to DNA as genetic storage. We have also added a discussion of possible mechanisms by which acetylation within the H3–H4 tetramer may be inherited across the cell cycle (lines 360–371).

[Revised text in Discussion]

To ensure the self-perpetuation of Kac in the H3-H4 tetramer, it would be important for p300/CBP to preferentially bind to nucleosomes containing H4K12ac/K16ac ($K_D = 25 \mu\text{M}$), H4K8ac/K12ac ($K_D = 90 \mu\text{M}$), or H3K14ac/K18ac ($K_D = 104 \mu\text{M}$) *via* bromodomain². Indeed, acetylation of these five residues (*i.e.*, H4K8ac, H4K12ac, H4K16ac, H3K14ac, and H3K18ac) is stably maintained during mitosis in mammalian cells³, suggesting that they are already acetylated in the early G₁ phase just after mitosis. For H4NTac in particular, H4K8ac is a mitotic bookmark for GAGA pioneer factor to activate its target genes in fly⁴. H4K12ac, which is recognized by the bromodomain pocket of p300_{BRPH}, is already acetylated in the cytoplasm^{5, 6, 7} and maintained during mitosis in living cells⁸. Also, H4K16ac is intergenerationally maintained in fly and mammals, making it instructive for future gene activation⁹. However, our data and those of another group¹⁰ show that p300 can hardly acetylate nucleosomal H4 at K16 and not much at K12. Given the distance between the H4NT pair, these data suggest that H4K8ac/K12ac self-perpetuates by p300/CBP with the help of H3K14ac/K18ac and cytoplasmic acetylation of H4K12. Self-perpetuation of H4K16ac presumably requires the MYST (MOZ, YBF2, SAS2, and Tip60)-type histone acetyltransferases, such as MOF/KAT8^{11, 12}. For H3NTac, the distance between the H3NT pair suggests that H3K14ac/K18ac may self-perpetuate by p300/CBP alone (Supplementary Fig. 18b). Such p300/CBP-mediated self-perpetuation of Kac in the H3-H4 tetramer would contribute to both the inheritance of Kac information to daughter cells and the p300/CBP-driven rapid transcriptional activation of specific genes immediately after mitosis¹³.

How is acetylation that is propagated within the H3-H4 tetramer inherited across the cell cycle? Individual histone octamers segregate conservatively²⁸ during DNA replication and are redeposited at the same position in one of the two daughter DNA molecules²⁹. Intriguingly, the H3-H4 tetramer partially splits into H3-H4 dimers³⁰ at transcriptionally active genes and enhancers^{31, 32}, possibly with the help of the histone chaperone ASF1^{33, 34, 35}.

Therefore, the acetylation information of the parental histones may be partially inherited by the H3-H4 tetramer, consisting of the parental H3-H4 dimer and a new H3-H4 dimer, in both transcriptionally active daughter chromatin regions. Then, p300/CBP may restore the diluted acetylation information derived from the parental H3-H4 dimer (Supplementary Fig. 18a), some of which may escape mitotic global histone deacetylation³⁶ and be inherited by both daughter chromatin. Mechanistically, p300/CBP-driven NTac and/or H3K56ac³⁷ may promote ASF1-dependent H3-H4 tetramer splitting and its semi-conservative redeposition.

I like the model presented on page 10, line 298ff. Could the authors draw a model to illustrate this?

[Response]

In line with your valuable suggestion, we have added a schematic diagram (new Fig. 5b) to illustrate our model (lines 372–397 and 409–425 corresponding to lines 298–313 and 325–341 of the original manuscript, respectively). In addition, one summary sentence has been added to the final paragraph of the Discussion (lines 432–437).

[Revised text in Discussion]

In other words, the essence of gene regulation unique to eukaryotes would be that the nucleosome has one [H3-H4]₂ tetramer in the inner core that inherits epigenetic information and two [H2A-H2B] dimers in the outer shell that express that information (Fig. 5b). Lysine acetylation in the nucleosome would have a duality of epigenetic inheritance and/or expression, depending on which histone it is in.

In addition to the above revisions in line with the reviewers' comments, we have made the following revisions.

- Two scientists who performed mass spectrometric analysis have been added as co-authors (lines 4, 10–11, and 1041–1042).
- One scientist who provided advice has been added to the Acknowledgments (line 1026).
- For readability, a biochemical summary and introduction to our model have been added at the end of the Introduction (lines 67–71).

[Revised text in Introduction]

In addition, using various pre-acetylated nucleosomes, we report the direction in which the catalytically active p300 multidomain 'reads/writes' histone acetylation within a single nucleosome. Based on our data, we propose a model in which p300/CBP ensures epigenetic inheritance and expression *via* intranucleosomal acetylation of the H3-H4 tetramer and H2B-H2A dimers, respectively.

- To accurately describe our results including the newly obtained mass spectrometry data, we have revised the following text as shown in red (lines 112–113).

[Revised text in Results]

These results suggest that when p300_{BRPHZT} 'reads' H4NTac at K12/K16, it 'writes' Kac primarily on H2BNT and then H3NT.

- The values of standard errors in Supplementary Tables 2 and 3 have been added (lines 237–242 and 291–296).
- To smoothly connect the logic of the background and results of this study, one sentence has been added at the beginning of the Discussion (lines 315–318).

[Revised text in Discussion]

Among histone acetyltransferases, p300/CBP rapidly activates the transcription of specific enhancer-regulated genes by binding to various DNA sequence-binding transcription factors (TFs) and through its own acetyltransferase activity³⁸, which is important for the regulation of diverse cellular responses and diseases.

- For consistency, words written in British English have been corrected to American English (lines 326 and 384).
- The description of 'two KK sequences' in the Discussion has been changed to 'multiple KK sequences' (line 402).
- The name of our model shown in the caption of Figure 5c (former Figure 4e) has been added to the final paragraph of the Discussion for clarity (line 426).
- Information on data availability has been added (lines 682–690).
- As background for the present study, references 6 (Brownell *et al. Cell* 84, 843, 1996), 7 (Shahbazian and Grunstein *Annu. Rev. Biochem.* 76, 75, 2007), 18 (Chen *et al. Nat. Struct.*

Mol. Biol. 21, 981, 2014), 20 (Narita *et al. BioRxiv*, 500456, 2022), 62 (Ferreon *et al. Proc. Natl. Acad. Sci. USA* 106, 13260, 2009), 63 (Feng *et al. Structure* 17, 202, 2009), 64 (Wojciak *et al. EMBO J.* 28, 948, 2009), 65 (Ibrahim *et al. Nat. Commun.* 13, 7759, 2022), and 74 (Davis *et al. Cell* 51, 987, 1987) have been added.

- Supplementary Figures 13 and 19 have been added to show source data for binding assays.

- Supplementary Figure 21 has been added to show source data for all cropped gel images.

References

1. Hatazawa S, *et al.* Structural basis for binding diversity of acetyltransferase p300 to the nucleosome. *iScience* **25**, 104563 (2022).
2. Ortega E, *et al.* Transcription factor dimerization activates the p300 acetyltransferase. *Nature* **562**, 538-544 (2018).
3. Behera V, *et al.* Interrogating Histone Acetylation and BRD4 as Mitotic Bookmarks of Transcription. *Cell Rep* **27**, 400-415 e405 (2019).
4. Bellec M, *et al.* The control of transcriptional memory by stable mitotic bookmarking. *Nat Commun* **13**, 1176 (2022).
5. Sobel RE, Cook RG, Perry CA, Annunziato AT, Allis CD. Conservation of deposition-related acetylation sites in newly synthesized histones H3 and H4. *Proc Natl Acad Sci U S A* **92**, 1237-1241 (1995).
6. Kleff S, Andrulis ED, Anderson CW, Sternglanz R. Identification of a gene encoding a yeast histone H4 acetyltransferase. *J Biol Chem* **270**, 24674-24677 (1995).
7. Parthun MR, Widom J, Gottschling DE. The major cytoplasmic histone acetyltransferase in yeast: links to chromatin replication and histone metabolism. *Cell* **87**, 85-94 (1996).
8. Ito T, *et al.* Real-time imaging of histone H4K12-specific acetylation determines the modes of action of histone deacetylase and bromodomain inhibitors. *Chem Biol* **18**, 495-507 (2011).
9. Samata M, *et al.* Intergenerationally Maintained Histone H4 Lysine 16 Acetylation Is Instructive for Future Gene Activation. *Cell* **182**, 127-144 e123 (2020).
10. Schiltz RL, Mizzen CA, Vassilev A, Cook RG, Allis CD, Nakatani Y. Overlapping but distinct patterns of histone acetylation by the human coactivators p300 and PCAF within nucleosomal substrates. *J Biol Chem* **274**, 1189-1192 (1999).
11. Smith ER, *et al.* The drosophila MSL complex acetylates histone H4 at lysine 16, a chromatin modification linked to dosage compensation. *Mol Cell Biol* **20**, 312-318 (2000).
12. Akhtar A, Becker PB. Activation of transcription through histone H4 acetylation by MOF, an acetyltransferase essential for dosage compensation in *Drosophila*. *Mol Cell* **5**, 367-375 (2000).
13. Pelham-Webb B, *et al.* H3K27ac bookmarking promotes rapid post-mitotic activation of the pluripotent stem cell program without impacting 3D chromatin reorganization. *Mol Cell* **81**, 1732-1748 e1738 (2021).
14. Ito T, Ikehara T, Nakagawa T, Kraus WL, Muramatsu M. p300-mediated acetylation facilitates the transfer of histone H2A-H2B dimers from nucleosomes to a histone chaperone. *Genes Dev* **14**, 1899-1907 (2000).
15. Puerta C, Hernandez F, Lopez-Alarcon L, Palacian E. Acetylation of histone H2A.H2B dimers facilitates transcription. *Biochem Biophys Res Commun* **210**, 409-416 (1995).
16. Kimura H, Cook PR. Kinetics of core histones in living human cells: little exchange of H3 and H4 and some rapid exchange of H2B. *J Cell Biol* **153**, 1341-1353 (2001).
17. Myers FA, Chong W, Evans DR, Thorne AW, Crane-Robinson C. Acetylation of histone H2B mirrors that of H4 and H3 at the chicken beta-globin locus but not at housekeeping genes. *J Biol Chem* **278**, 36315-36322 (2003).

18. Narita T, Higashijima Y, Kilic S, Maskey E, Neumann K, Choudhary C. The logic of native enhancer-promoter compatibility and cell-type-specific gene expression variation. *bioRxiv*, 2022.2007.2018.500456 (2022).
19. Narita T, Higashijima Y, Kilic S, Liebner T, Walter J, Choudhary C. A unique H2B acetylation signature marks active enhancers and predicts their target genes. *bioRxiv*, 2022.2007.2018.500459 (2022).
20. Ehara H, Kujirai T, Shirouzu M, Kurumizaka H, Sekine SI. Structural basis of nucleosome disassembly and reassembly by RNAPII elongation complex with FACT. *Science*, eabp9466 (2022).
21. Hsu CC, *et al.* Recognition of histone acetylation by the GAS41 YEATS domain promotes H2A.Z deposition in non-small cell lung cancer. *Genes Dev* **32**, 58-69 (2018).
22. Colino-Sanguino Y, *et al.* A Read/Write Mechanism Connects p300 Bromodomain Function to H2A.Z Acetylation. *iScience* **21**, 773-788 (2019).
23. Zhang Y, *et al.* The ZZ domain of p300 mediates specificity of the adjacent HAT domain for histone H3. *Nat Struct Mol Biol* **25**, 841-849 (2018).
24. Hamamori Y, *et al.* Regulation of histone acetyltransferases p300 and PCAF by the bHLH protein twist and adenoviral oncoprotein E1A. *Cell* **96**, 405-413 (1999).
25. Thompson PR, *et al.* Regulation of the p300 HAT domain via a novel activation loop. *Nat Struct Mol Biol* **11**, 308-315 (2004).
26. Karanam B, Jiang L, Wang L, Kelleher NL, Cole PA. Kinetic and mass spectrometric analysis of p300 histone acetyltransferase domain autoacetylation. *J Biol Chem* **281**, 40292-40301 (2006).
27. Kaczmarek Z, *et al.* Structure of p300 in complex with acyl-CoA variants. *Nat Chem Biol* **13**, 21-29 (2017).
28. Leffak IM, Grainger R, Weintraub H. Conservative assembly and segregation of nucleosomal histones. *Cell* **12**, 837-845 (1977).
29. Schlissel G, Rine J. The nucleosome core particle remembers its position through DNA replication and RNA transcription. *Proc Natl Acad Sci U S A* **116**, 20605-20611 (2019).
30. Xu M, Long C, Chen X, Huang C, Chen S, Zhu B. Partitioning of histone H3-H4 tetramers during DNA replication-dependent chromatin assembly. *Science* **328**, 94-98 (2010).
31. Katan-Khaykovich Y, Struhl K. Splitting of H3-H4 tetramers at transcriptionally active genes undergoing dynamic histone exchange. *Proc Natl Acad Sci U S A* **108**, 1296-1301 (2011).
32. Huang C, *et al.* H3.3-H4 tetramer splitting events feature cell-type specific enhancers. *PLoS Genet* **9**, e1003558 (2013).
33. Tagami H, Ray-Gallet D, Almouzni G, Nakatani Y. Histone H3.1 and H3.3 complexes mediate nucleosome assembly pathways dependent or independent of DNA synthesis. *Cell* **116**, 51-61 (2004).
34. Zhu B, Reinberg D. Epigenetic inheritance: uncontested? *Cell Res* **21**, 435-441 (2011).
35. Budhavarapu VN, Chavez M, Tyler JK. How is epigenetic information maintained through DNA replication? *Epigenetics Chromatin* **6**, 32 (2013).
36. Schneider MWG, *et al.* A mitotic chromatin phase transition prevents perforation by microtubules. *Nature* **609**, 183-190 (2022).
37. Das C, Lucia MS, Hansen KC, Tyler JK. CBP/p300-mediated acetylation of histone H3 on lysine 56. *Nature* **459**, 113-117 (2009).
38. Narita T, *et al.* Enhancers are activated by p300/CBP activity-dependent PIC assembly, RNAPII recruitment, and pause release. *Mol Cell* **81**, 2166-2182 e2166 (2021).

REVIEWERS' COMMENTS

Reviewer #1 (Remarks to the Author):

The authors have adequately addressed my concerns.

Reviewer #2 (Remarks to the Author):

The authors have done a good job with the revision. The data in supplementary table 2 is very difficult to look at. Could this be compiled in a chart?

Response to the reviewers' comments

To Reviewer 1

[Comments]

The authors have adequately addressed my concerns.

[Response]

We appreciate this reviewer for their favorable evaluation.

To Reviewer 2

[Comments]

The authors have done a good job with the revision. The data in supplementary table 2 is very difficult to look at. Could this be compiled in a chart?

[Response]

We appreciate this reviewer's valuable suggestion. A chart corresponding to Supplementary Table 2 has been added as Supplementary Figure 3.

In addition to the above revisions in line with the reviewers' comments, we have made the following revisions.

- One scientist who assisted in sample preparation has been added to the Acknowledgments (line 1174).
- References 62 and 63 have been added for discussion of this study.
- Following the instructions in the author checklist, we have revised some of the descriptions in the Methods as highlighted in red. Accordingly, references 78, 90, and 91 have been added and previous references 77 and 94 have been deleted.